ecology/evolution/biogeography

assymetrical migration, Antarctic Peninsula Coastal Current, NexTRAD

**Authors for correspondence:**
Carlos P. Muñoz-Ramírez
e-mail: carlos.munoz@umce.cl
Antonio Brante
e-mail: abrante@ucsc.cl

# Gene flow in the Antarctic bivalve *Aequiyoldia eightsii* (Jay, 1839) suggests a role for the Antarctic Peninsula Coastal Current in larval dispersal

Carlos P. Muñoz-Ramírez[1,2,3], David K. A. Barnes[4], Leyla Cárdenas[5], Michael P. Meredith[4], Simon A. Morley[4], Alejandro Roman-Gonzalez[6], Chester J. Sands[4], James Scourse[6] and Antonio Brante[2,3]

[1]Instituto de Entomología, Facultad de Ciencias Básicas, Universidad Metropolitana de Ciencias de la Educación, Santiago, Chile
[2]Facultad de Ciencias, Universidad Católica de la Santísima Concepción, Concepción, Chile
[3]Centro de Investigación en Biodiversidad y Ambientes Sustentables (CIBAS), Universidad Católica de la Santísima Concepción, Concepción, Chile
[4]British Antarctic Survey, Natural Environment Research Council, Cambridge, UK
[5]Centro FONDAP de Investigación Dinámica de Ecosistemas Marinos de Altas Latitudes (IDEAL), Instituto de Ciencias Ambientales y Evolutivas, Facultad de Ciencias, UniversidadAustral de Chile, Valdivia, Chile
[6]College of Life and Environmental Sciences, University of Exeter, Penryn, Cornwall TR10 9EZ, UK

CPM-R, 0000-0003-1348-5476; DKAB, 0000-0002-9076-7867;
LC, 0000-0003-0676-6704; MPM, 0000-0002-7342-7756;
SAM, 0000-0002-7761-660X; AR-G, 0000-0002-0801-3768;
CJS, 0000-0003-1028-0328; JS, 0000-0003-2658-8730;
AB, 0000-0002-2699-9700

The Antarctic Circumpolar Current (ACC) dominates the open-ocean circulation of the Southern Ocean, and both isolates and connects the Southern Ocean biodiversity. However, the impact on biological processes of other Southern Ocean currents is less clear. Adjacent to the West Antarctic Peninsula (WAP), the ACC flows offshore in a northeastward direction, whereas the Antarctic Peninsula Coastal Current (APCC) follows a complex circulation pattern along the coast, with topographically influenced deflections depending on the area. Using genomic data, we estimated genetic structure and migration rates between populations of the benthic bivalve *Aequiyoldia eightsii*

from the shallows of southern South America and the WAP to test the role of the ACC and the APCC in its dispersal. We found strong genetic structure across the ACC (between southern South America and Antarctica) and moderate structure between populations of the WAP. Migration rates along the WAP were consistent with the APCC being important for species dispersal. Along with supporting current knowledge about ocean circulation models at the WAP, migration from the tip of the Antarctic Peninsula to the Bellingshausen Sea highlights the complexities of Southern Ocean circulation. This study provides novel biological evidence of a role of the APCC as a driver of species dispersal and highlights the power of genomic data for aiding in the understanding of the influence of complex oceanographic processes in shaping the population structure of marine species.

# 1. Introduction

It is well established that Antarctic biodiversity has been strongly influenced by oceanic currents [1]. The formation of the Antarctic Circumpolar Current (ACC) some approximately 30 Ma, flowing clockwise around Antarctica, has been regarded as one of the main events driving the isolation of local marine fauna [2]. The absence of sharks, rarity of other groups like rays and crabs, and the increased diversity of yet others, like pycnogona, polychaete worms and peracarid crustaceans exemplify the isolation and uniqueness of Antarctic biodiversity [3]. Along with serving as an isolating force of species into the Southern Ocean [4], the ACC has been regarded as an effective dispersal agent within the Southern Ocean, facilitating migration around Antarctica. Large circumpolar ranges and patterns of genetic connectivity of several taxa including the limpet *Nacella concinna* [5], the octopus *Pareledone aequipapillae* [6] and the crinoid *Promachocrinus kerguelensis* [7] are only a few of the examples supporting the role of the ACC as a driver of connectivity of marine fauna within the Southern Ocean. Comparatively little is known, however, about other Antarctic currents and their potential role on the dispersal of benthic species.

In contrast to the ACC, the Antarctic Peninsula Coastal Current (APCC), located in the Western Antarctic Peninsula (WAP), follows a complex circulation pattern along the coast and around the many islands of the WAP, forced by freshwater discharge and downwelling-favourable winds near the coast [8]. Although many aspects of the APCC circulation remain undetermined, present understanding is that it consists of generally southward flows alongside the large Adelaide and Alexander Islands, with a presumed cyclonic circulation within Marguerite Bay (figure 1). Southward flow from Anvers Island towards Adelaide Island inshore of the smaller islands present is inferred, and there are observations of a generally northward flow north of Anvers Island toward Bransfield Strait. It is stressed that these patterns are based on sparse data available, and temporal variability in the APCC flow is not yet well resolved. The role of the APCC in the dispersal ecology of benthic Antarctic fauna remains unstudied, but its prevalence during the warmer season, when several taxa reproduce and spawn [10], means this current might be a relevant environmental force for larval dispersal and population connectivity.

The bivalve *Aequiyoldia eightsii* is one of the most abundant benthic species from the shallows of the Antarctic Peninsula [11]. It reproduces during April–May, producing what are thought to be lecithotrophic larvae [10], a type of non-feeding larvae with a short planktonic stage. Given this planktonic stage, the species' dispersal potential is predicted to be largely mediated by ocean currents [12]. The species has also been documented in waters around Southern America, across the ACC, but recent molecular work suggests this might be an undescribed species, despite the absence of obvious macro-morphological differences [13]. While the strong open-ocean flow of the ACC has been recognized as an important potential dispersal mechanism in the Southern Ocean, coastal currents such as the APCC could be more relevant for dispersal of benthic species such as *A. eightsii* in shallow waters.

Genome-wide molecular data such as that obtained from restriction site associated DNA sequencing (RADSeq) have the power to estimate structure and patterns of gene flow at a range of different spatial scales providing the means to detect fine scale genetic subdivision and assess the magnitude and direction of ongoing gene flow [14]. Understanding gene flow and dispersal in Antarctic organisms is important because it contributes to a better understanding of genetic structure, metapopulation dynamics [15] and resilience [16], all important in the context of understanding the impact of global climate change and a warming Antarctica [17]. For instance, understanding which populations could behave as source populations and which as sink populations would not only contribute to a better understanding of the oceanographic processes that drive these dynamics, but also provide valuable information for population and species conservation. In this context, knowing which populations

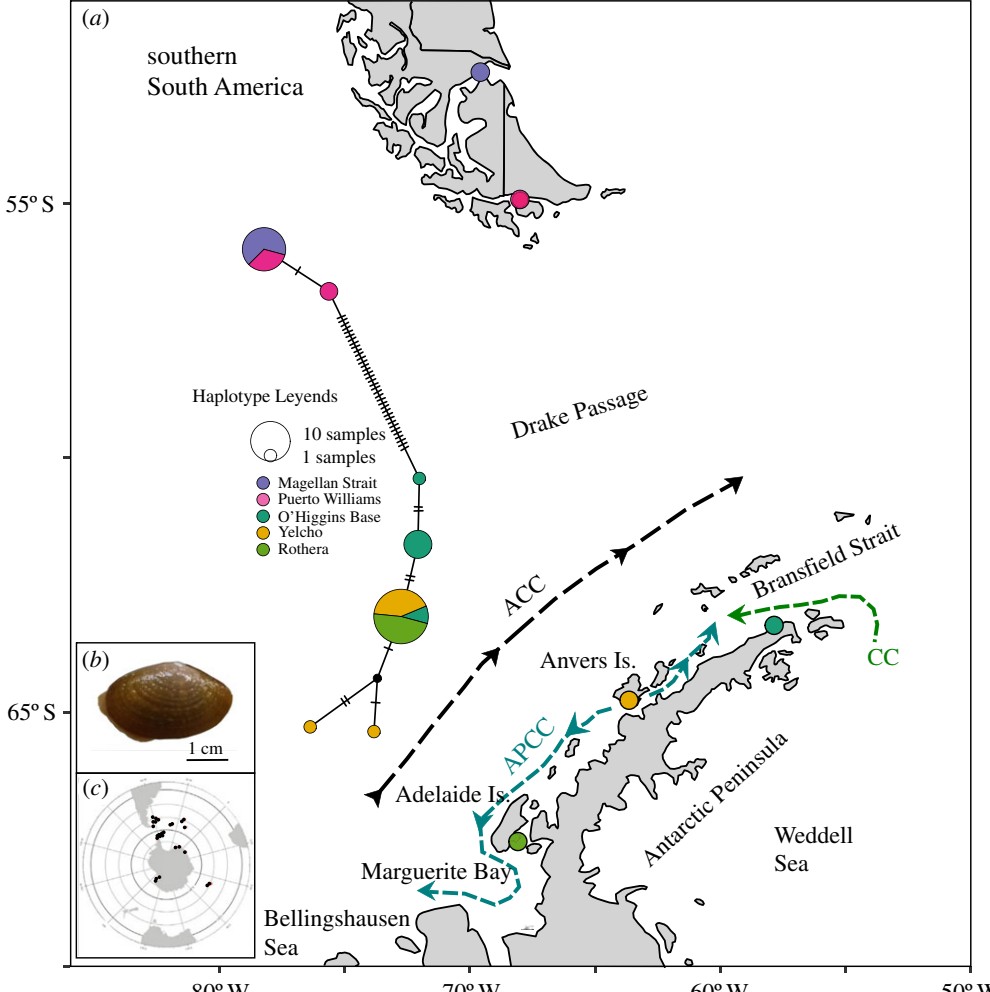

**Figure 1.** (*a*) Study area and sampling localities for the species *Aequiyoldia eightsii* including an haplotype network inferred from the mtDNA (COI) dataset and a simplified scheme of the ocean currents present in the Western Antarctic Penninsula (ACC, Antarctic Circumpolar Current; APCC, Antarctic Peninsula Coastal Current; CC, Antarctic Coastal Current (from [8])). Haplotype colours are matched with their corresponding geographical distribution. (*b*) Specimen of *Aequiyoldia eightsii* from Yelcho. (*c*) Geographical distribution of the genus *Aequiyoldia* [9].

behave as source populations will help focus conservation efforts on specific areas, making management plans more tractable. In the context of a global climate change, identifying the currents that contribute to population connectivity may help predict future connectivity patterns through understanding of how climate change may influence currents and circulation patterns.

In this study, we use genomic data to (i) evaluate genetic structure across the ACC, i.e. between South America and Antarctica, and within both these regions, and (ii) estimate patterns of directional gene flow to test the role of ocean currents (ACC versus APCC) on species dispersal along the WAP.

# 2. Material and methods

## 2.1. Sampling and molecular data collection

*Aequiyoldia eightsii* is widely distributed in the Southern Ocean, but owing to sampling limitations and the need for recently collected specimens, this study focused on the southern South America and the WAP regions. Forty-one individuals of *A. eightsii* were collected from five sites, two from southern South America and three from the WAP (figure 1 and table 1). Genomic DNA was extracted with the Qiagen DNeasy Blood & Tissue kit, following the manufacturer's protocol. Mitochondrial DNA (mtDNA) data were collected to conduct a preliminary assessment of phylogeographic patterns and to

**Table 1.** Sampling localities for *Aequiyoldia eightsii*. (*n* = Number of individuals sampled. Last three columns show the number of loci obtained from the NextRAD protocol when a minimum of four (M4), 10 (M10) or 20 (M20) individuals are required for a locus to be included. Loci count are average per population. SA, South America; WAP, Western Antarctic Peninsula.)

| site | region | latitude | longitude | *n* | M4 | M10 | M20 |
|------|--------|----------|-----------|-----|-----|------|------|
| Magellan Strait | SA | −52.4231 | −69.5717 | 8 | 87 067 | 35 046 | 729 |
| Puerto Williams | SA | −54.9258 | −67.9911 | 6 | 39 621 | 16 944 | 558 |
| O'Higgins Base | WAP | −63.3208 | −57.8986 | 8 | 39 615 | 16 753 | 440 |
| Yelcho | WAP | −64.8758 | −63.6089 | 10 | 22 077 | 10 146 | 224 |
| Rothera | WAP | −67.5709 | −68.1269 | 9 | 104 922 | 42 349 | 746 |

confirm a single evolutionary lineage, given that multiple lineages have been documented previously [13]. A 629 bp fragment of the cytochrome *c* oxidase subunit I (COI) gene was amplified using universal primers from [18] with final concentrations for polymerase chain reaction components per 25 µl reaction as follows: 25 ng template DNA, 0.25 µM of each primer, 0.625 units of GoTaq DNA polymerase (Promega, Madison, WI, USA), 0.1 mM of each dNTP, 2.5 µl of 10 reaction buffer and 2.5 mM $MgCl_2$. Amplification parameters were as follows: 95°C for 2 min followed by 35 cycles of 95°C for 30 s, 48°C for 30 s, 72°C for 90 s, and a final extension of 72°C for 7 min. Purification and sequencing were conducted at MACROGEN Inc. (South Korea). Chromatograms were edited in CODONCODE ALIGNER 8.0.2 (Dedham, MA, USA). Sequences were imported to BIOEDIT 7.0.5.2 [19], aligned using the CLUSTAL W algorithm, available within BIOEDIT, and checked by eye. All sequences were deposited in GenBank (accession numbers MT176643–MT176683).

Genome-wide data were obtained through the Nextera-tagmented reductively amplified DNA protocol (NextRAD; SNPSaurus LLC, Eugene Oregon) of [20] to improve genetic resolution at the spatial scale analysed. Genomic DNA was converted into NextRAD genotyping-by-sequencing libraries. Genomic DNA was first fragmented with Nextera DNA FLex reagent (Illumina, Inc), which also ligates short adapter sequences to the ends of the fragments. The Nextera reaction was scaled for fragmenting 24 ng of genomic DNA. Fragmented DNA was then amplified for 27 cycles at 74°C, with one of the primers matching the adapter and extending 10 nucleotides into the genomic DNA with the selective sequence GTGTAGAGCC. Thus, only fragments starting with a sequence that can be hybridized by the selective sequence of the primer will be efficiently amplified. The NextRAD libraries were sequenced on a HiSeq 4000 with one lane of 150 bp reads (University of Oregon). Reads obtained from the NextRAD protocol were processed using the ipyRAD pipeline ver. 7.30 [21]. Base calls with a quality score of less than 20 were converted into Ns, and any read with greater than 5 Ns was discarded. Illumina adaptors and restriction sequences were removed during filtering. Filtered reads within a sample were clustered using a threshold of 90%. Error rate and heterozygosity were estimated from the loci clusters for each individual, and the averages were used to establish consensus sequences. Clusters with a sequencing depth of less than 6 were discarded, and only clusters with two alleles (to avoid potential paralogous loci) were retained. Consensus loci built within samples were subsequently clustered among samples using a similarity threshold of 90% and then aligned (a maximum of eight indels allowed). Loci with heterozygous alleles shared across more than 50% of individuals were also discarded. Depending on the type of analysis and their sensitivity to missing data, different genomic datasets were produced to maximize the amount of genetic information available for each analysis by changing the levels of coverage/missing data allowed (changing the minimum number of samples per locus for output in ipyRAD's step 7). The electronic supplementary material, table S1 shows summary statistics for data generated by setting the minimum number of individuals per locus to approximately 10% (four individuals), approximately 25% (10 individuals) and approximately 50% (20 individuals).

## 2.2. Molecular analyses

To estimate levels of population differentiation for the COI data, genetic distance (*p*-distance) between populations were calculated using the R-package ADEGENET [22]. To visually display relationships between COI haplotypes, a haplotype network was estimated using the median-joining network

algorithm of [23] implemented in the program POPART [24]. In addition, COI structure was tested calculating pairwise $F$st values using ADEGENET.

The genomic data generated by NextRAD were used to infer genetic structure through several methods. First, relationships between each individual were estimated by building a maximum likelihood (ML) tree using the program RAxML v. 8.1.16 [25] with a concatenated dataset consisting of 44 178 loci, each loci including the full sequence with variable (single nucleotide polymorphisms (SNPs)) and invariable sites (matrix length = 5 973 095 bp). This matrix included all loci that were present in at least four individuals (approx. 10% of coverage). Including loci with missing data has been advised for phylogenetic analyses because this strategy not only increases the amount of information available for inference (i.e. by increasing the number of loci in the matrix), but also because it prevents sampling biases—excluding missing data reduces the mutational spectrum represented in the sample by disproportionately excluding loci with the highest mutation rates [26]. Given that we used the full sequence of each locus (not only the variable sites), the GTR + $\Gamma$ model of nucleotide evolution was used as recommended in the software manual [25]. Support was assessed by 200 nonparametric bootstrap replicates, followed by a search for the best-scoring ML tree.

Subsequently, the software STRUCTURE 2.3.4 [27], that performs a model-based inference of population subdivision, was used to evaluate population groups using an SNP data matrix of 7781 (dataset including loci present in at least 10 individuals). When more than one SNP are present at a locus, the ipyRAD program choses one of them randomly to ensure the independence of each locus. Number of populations to be tested ($K$-values) ranged from 1 to 5. Ten independent runs per $K$ were conducted, each with 100 000 burn-in and 300 000 Markov chain Monte Carlo (MCMC) iterations, using the 'Admixture Model' and 'Correlated Allele Frequency Model' with default settings, except for the Lambda parameter, which was empirically estimated in a preliminary run. Results were not different using more burn-in or MCMC iterations. STRUCTURE HARVESTER [28] and DISTRUCT [29] were used to visualize results, and the most probable $K$ was chosen based on $\Delta K$ [30]. After a first round of analyses was performed, subsequent analyses were conducted on each of the resulting clusters to determine whether any level of substructure could be identified. In addition to the STRUCTURE analyses, a discriminant analysis of principal components (DAPC) was conducted using the R-package ADEGENET [22]. This analysis transforms the data using a principal component analysis (PCA) followed by a discriminant analysis on the uncorrelated PCA variables to produce synthetic discriminant functions or axes that minimize within-group variation while maximizing between-group variation [31]. First, the number of clusters was explored using the find.clusters function, which uses a sequential $K$-means clustering algorithm and Bayesian Information Criterion to select the best-fit model (number of clusters) [31]. Subsequently, the DAPC analysis was conducted using the estimated $K$-value, and the actual locations (five) as groupings for comparison, retaining in each case two principal components. A dataset using 811 SNPs, which contained loci that were present in at least 20 individuals, was used for the DAPC analysis (see the electronic supplementary material for further details). We complemented these cluster analyses by also calculating pairwise $F$st values between sites using the same dataset used for the STRUCTURE analysis and the R-package STAMPP [32].

Given that *A. eightsii* has a pelagic life stage, and its dispersal is likely to be influenced by ocean currents, rates of migration were tested using divMIGRATE-online [33] and BAYESASS3-SNPS [34]. The divMIGRATE-online method is based on defining a hypothetical pool of migrants for a given pair of populations and estimating an appropriate measure of genetic differentiation between each of the two populations and the hypothetical pool. The directional genetic differentiation can then be used to estimate the relative levels of migration between the two populations where the larger of the two relative values indicates the population that is probably a source population, while the smaller of the two estimates indicates the sink population. The $D$ distance [35] was used as the measure of genetic differentiation and 1000 bootstrap replicates were used for statistical testing of asymmetrical migration. The input dataset for this analysis was the same we used for the STRUCTURE analysis, but converted to a genepop input file using ADEGENET [22] (see further methodological details in the electronic supplementary material). A general southwest-northeast direction of gene flow (Rothera to Yelcho and Yelcho to O'Higgins) would support ACC importance for larval dispersal, whereas a Yelcho to Rothera direction and a Yelcho to O'Higgins direction would support a greater influence of the APCC as the main agent of dispersal in the WAP. BAYESASS3-SNPS [34], an expansion of BAYESASS [36] that facilitates the use of large amounts of SNPs and automates the fine tuning of model parameters, was also used to estimate migration rates between populations. A first round of analyses was conducted with the auto-tuning tool developed as part of BAYESASS3-SNPS to extensively calibrate parameters. Subsequently, the final analysis was conducted by running five replicates for 5 000 000 MCMC generations, sampling every 500 generations and discarding

the first 500 000 generations as burn-in. Convergence of parameter estimations were checked with TRACER **6**
v1.7.1 [37]. Data and input files for analyses, as well as further methodological details, are available at [38].

# 3. Results

## 3.1. Genetic patterns from the mitochondrial DNA COI data

Genetic differentiation in *A. eightsii* was high between South America and Antarctica. The average number of nucleotide differences between these two regions was 38.93 (6.19%), while the average number of nucleotide differences within regions was 1.78 (0.28%) and 2.28 (0.36%) for South America and Antarctica, respectively. The haplotype network confirmed the high genetic divergence between regions (figure 1), and showed some, although much lower, genetic structure between sites from the WAP. At the WAP, although only one main haplogroup was found (ruling out the presence of more than a single lineage within our samples), there was a high-frequency haplotype present at all WAP sites, plus a few private haplotypes from O'Higgins and Yelcho. Consequently, genetic structure based on $F$st (electronic supplementary material, table S2) showed high genetic differentiation between South America and Antarctica. Within regions, statistically significant differentiation between some Antarctic populations was also found. There was significant differentiation between O'Higgins Base and the other two Antarctic sites, Yelcho ($F$st = 0.360, $p$ = 0.012) and Rothera ($F$st = 0.469, $p$ = 0.008), but no significant differentiation was found between Yelcho and Rothera ($F$st = 0.072, $p$ = 0.304) nor between the two South American populations ($F$st = 0.142, $p$ = 0.131).

## 3.2. Genomic data and genetic structure

The total number of loci found by ipyRAD was 352 263. After the filtering process and quality controls, the total amount of loci was 297 417. As with many RADSeq datasets, which are characterized by moderate to high levels of missing data [26,39,40], most of these loci were present only for a few individuals (for instance, 253 239 were present at three or fewer samples). Subsequently, datasets obtained with the constraint of having loci represented for at least 4, 10 and 20 individuals produced a total of 44 178, 8808 and 941 loci, respectively. Locus length varied from 16 bp to 266 bp, with a mean locus length of 135 bp. Of these loci, 5090 were invariable, whereas 39 088 were variable. The number of variable sites (SNPs) per locus varied from 1 to 19, with an average of 5.4 SNPs per locus.

ML phylogenetic relationships showed strong phylogeographic structure, with clades strongly reflecting geography (figure 2*a*). Support for the different clades varied from very high at the Rothera site (bootstrap support = 100) to weak in the Puerto Williams clade (bootstrap support = 53). Antarctica and South America were recovered as the two major clades, with all populations within these regions being recovered as reciprocally monophyletic with high support (bootstrap support = 100).

The STRUCTURE analyses confirmed the strong genetic differentiation between the South American and Antarctic regions found in the ML tree (figure 2*a*). For the entire dataset, including all samples in a single analysis, the highest supported $K$ (as suggested by the Evanno test) was $K$ = 2, matching exactly the Antarctic and South American regions. After rerunning STRUCTURE independently for each region, the results did not support genetic substructure in South America. On the other hand, the analysis for the Antarctic region supported the existence of two clusters ($K$ = 2) with varying degrees of admixture. One cluster was mainly associated with Rothera, while the second cluster was mainly present in Yelcho. The northernmost site, O'Higgins Base, presented individuals with different levels of admixture from both clusters.

The DAPC results were broadly consistent with the STRUCTURE results in that it clearly separated South America and Antarctica. However, there were some discrepancies regarding structure between sites at the WAP (figure 3*a,b*). The two first PC axes explained, cumulatively, 25.36% of the total variance, with the first, second and third PC axes explaining 21.91% and 3.45% of the total variance, respectively (electronic supplementary material, figures S1). At the WAP, Rothera was the most differentiated site, whereas the O'Higgins Base and Yelcho sites showed less differentiation, and a greater overlap in genetic space (figure 3*a*). The Magellan Strait and Puerto Williams sites in the South American region showed moderate differentiation. The Bayesian information criterion supported $K$ = 2 clusters as the optimal solution (electronic supplementary material, figure S2), consistent with a separation between the South American and Antarctic regions (figure 3*c*). Two individuals from the South American region showed admixed memberships (PW_04 and EM_06) with the Antarctic locality Yelcho, a result that may be influenced by a high level of missing data for these individuals

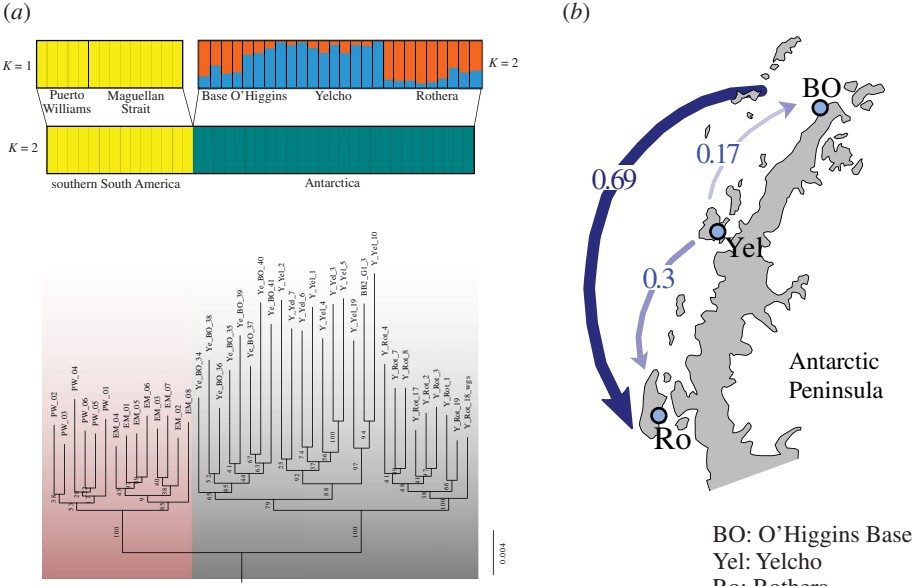

**Figure 2.** (*a*) Cluster analyses (up) and maximum-likelihood phylogenetic tree (down) for *Aequiyoldia eightsii* populations from Antarctica and South America using the NextRAD dataset. Top cluster plot represents independent analysis within regions. (*b*) Depiction of asymmetric gene flow between populations from the Western Antarctic Peninsula calculated with divMIGRATE-online using the *D* statistic of Jost [34] and 999 bootstrap permutations. Wider and bolder arrows represent stronger gene flow in the direction of the arrows. Graphics modified from original (electronic supplementary material, figure S1) for illustrative purposes.

which, coincidentally, presented the lowest number of loci in the dataset (electronic supplementary material table S1). *F*st values for the genomic dataset were concordant with the STRUCTURE analysis, showing high differentiation between South America and Antarctica, and also showing Yelcho as the most differentiated site at the WAP (table 2).

The analysis of directional gene flow (divMIGRATE-online) revealed an important asymmetry in the migration rates between populations in the WAP (figure 2*b*). A moderate, but significantly higher migration rate was detected from Yelcho to Rothera ($D = 0.30$) as well as from Yelcho to O'Higgins Base ($D = 0.17$) in agreement with a role for the APCC on larval dispersal. Strikingly, the highest rate of migration occurred from O'Higgins Base to Rothera ($D = 0.69$). High migration rates were inferred between the two South American sites, although no significant asymmetrical migration was found (electronic supplementary material, figure S3 and table S3).

The BAYESASS migration rate patterns were qualitatively similar to those of divMIGRATE-online. The mean migration rate from Yelcho to Rothera (0.024) was slightly higher than from Rothera to Yelcho (0.022) and the mean migration rate from Yelcho to O'Higgins Base (0.026) was slightly higher than from O'Higgins Base to Yelcho (0.022). However, confidence intervals were very wide indicating no statistical support for asymmetrical migration between any population pair (electronic supplementary material, table S4). Furthermore, difficulties to obtain adequate acceptance rates for parameter estimations and to achieving convergence for the Markov chains make these results hard to interpret.

## 4. Discussion

In this study, we aimed to investigate patterns of genetic structure for the Antarctic bivalve *A. eightsii* to assess the role of the ACC and the APCC on shaping connectivity patterns and genetic differentiation. Genome-wide markers have been recently used to support ongoing gene flow across the Drake Passage in the brittle star *Astrotoma agassizii* [41], in contrast to the evidence of no connectivity found earlier for the species based only on a few markers [42]. There is also recent evidence, based on SNP markers, that marine benthos (specifically brittle stars) have northwards gene flow across the ACC from Antarctica to South America [43]. Thus, different sources of genetic information can often provide new insights about the processes driving patterns of spatial genetic diversity. In this study, we found no evidence of significant migration between Antarctica and South American populations of the bivalve *A. eightsii* confirming the results found by previous studies based on fewer markers

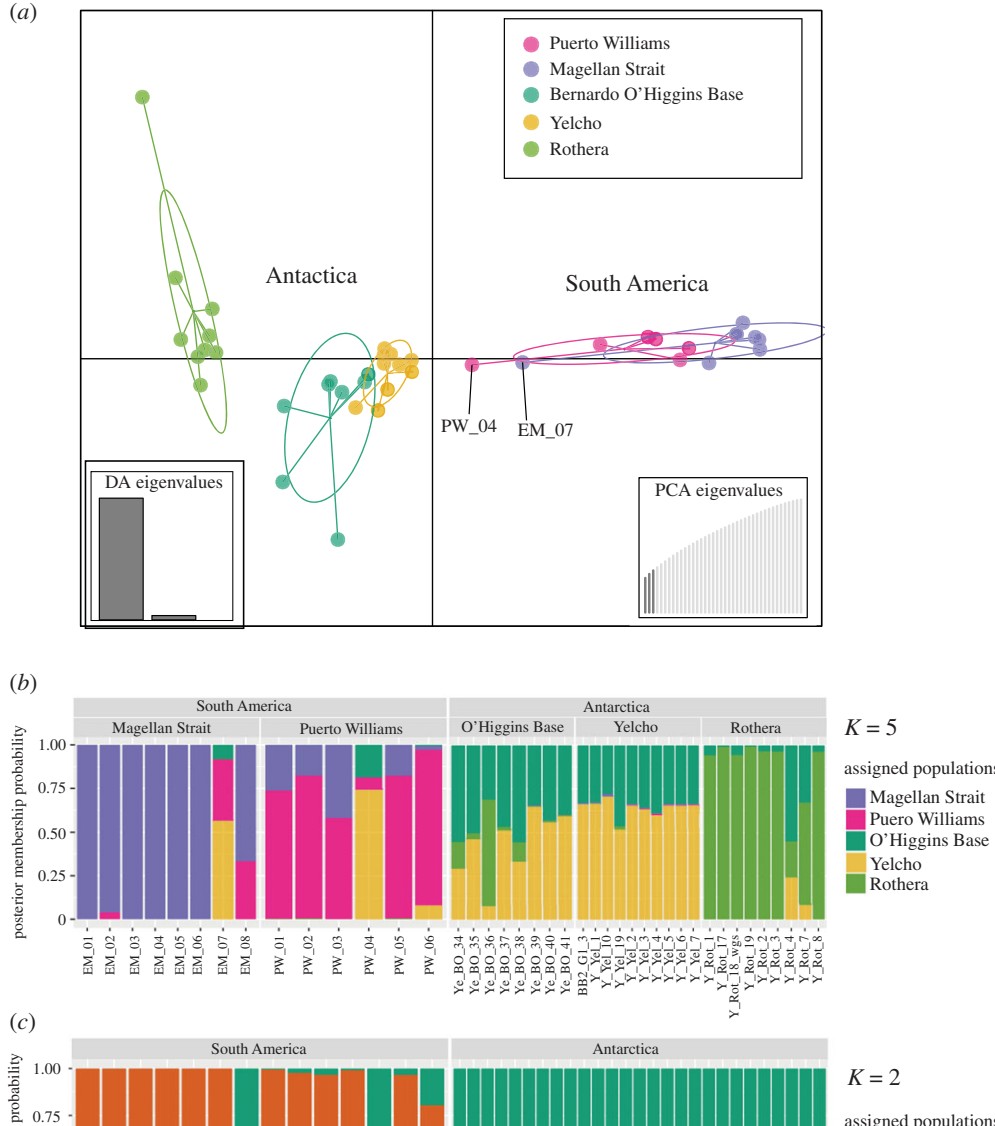

**Figure 3.** Discriminant analyses of principal components, DAPC, for individuals of the bivalve *Aequiyoldia eightsii* from South America and Antarctica based on single-nucleotide polymorphism data. (*a*) Discriminant analysis plot. (*b,c*), DAPC membership plots for *K* = 5 (*b*) and *K* = 2 (*c*) cluster assignments. DA, discriminant analysis. Samples with the lowest number of loci are labelled in the DAPC plot (*a*).

(e.g. [13,44]). Although some results from the DAPC analysis yielded conflicting results by showing two individuals from the South American region with membership probabilities mixed with Antarctic populations, all other analyses (RAxML phylogeny and STRUCTURE) showed a strong differentiation between these regions. Thus, it is likely that the conflicting evidence correspond to an artefact owing to high levels of missing data for some samples. Although the impact of missing data has not been thoroughly studied in the context of genetics, it is known that missing data can severely impact PCA performance, especially when missing data are not distributed randomly [45]. In fact, in our dataset the two individuals that presented the conflicting pattern (EM_07 and PW_04) were also the ones that had the fewest loci (electronic supplementary material, table S1), suggesting that the lack of

**Table 2.** *F*st values between sites for the bivalve *Aequiyoldia eightsii* (*n* = 41) based on 7973 SNPs and a minimum of 10 individuals with data for each loci.

|  | MS | PW | OH | Ye |
|---|---|---|---|---|
| Magellan Strait (MS) | — | — | — | — |
| Puerto Williams (PW) | 0.1266 | — | — | — |
| O'Higgins Base (OH) | 0.6339 | 0.5797 | — | — |
| Yelcho (Ye) | 0.6501 | 0.5599 | 0.1595 | — |
| Rothera (Ro) | 0.5914 | 0.5482 | 0.0893 | 0.2332 |

information in these individuals may have determined their position towards the centre of the PCA plot, placing them artificially close to the Antarctic cluster (electronic supplementary material, figure S1).

Within regions, genetic structure was more evident in the WAP. Both COI and genomic data evidenced some degree of genetic structure between populations, although some discrepancies between *F*st patterns from mtDNA and SNP, and STRUCTURE and DAPC results regarding which site was the most differentiated were evident. *F*st patterns between COI and SNPs datasets showed different levels of differentiation for Yelcho, being the mtDNA data unable to detect significant differentiation. This may be owing, at least in part, to the lower amount of information available in the single-gene COI data. Regarding differences between STRUCTURE and DAPC, Rothera was identified as the most distinct by DAPC, whereas Yelcho was the most distinct based on STRUCTURE. We are not certain about the reasons for the difference between analyses. Perhaps, it could be related with the underlying assumptions of each programme, as STRUCTURE assumes populations are at Hardy-Weinberg equilibrium [27], whereas DAPC is model free [31]. Alternatively, differences could also be associated with levels of missing data, DAPC being apparently more sensitive to high levels of missing data or missing data not being uniformly distributed across samples [45].

We found evidence of asymmetrical migration between Antarctic populations that was broadly consistent with the circulation patterns of the APCC (figures 1 and 2*b*). A predominant migration from Yelcho to O'Higgins Base and from Yelcho to Rothera matches the circulation of the APCC in this area, which is conceived to flow meridionally away around Yelcho (Anvers Island). From Anvers Island, one branch flows southwestward towards Margarite Bay, where Rothera Research Station is located, whereas the other branch flows northeastward towards the opposing flow of the Antarctic Coastal Current which flows around the tip of the Peninsula from the Weddell Sea through the Bransfield Strait [8]. More counterintuitive, however, is the asymmetrical migration inferred from O'Higgins Base to Rothera, because nearshore flow would be opposed by the general northeast flow of the currents north of Anvers Island. However, and as noted above, currents in this area are spatially complex and strongly structured by topographic steering and the presence of many small islands, and most models depicting flow directions are simplifications of this complexity [8,46]. Renner *et al*. [45,46], using an ice-ocean modelling approach and empirical data from near-surface drifters released in the Weddell Sea near the tip of the Antarctic Peninsula, found that although most of connectivity goes northeast across the Scotia Sea towards South Georgia, a significant portion goes initially westward around the tip of the Antarctic Peninsula and then southward to the Bellingshausen Sea. This ocean circulation fits well with the gene flow pattern observed in *A. eightsii* because it would mean that larval transportation from the tip of the Antarctic Peninsula to Rothera can occur following this southward current at some distance from the coast, avoiding the near-coast, northeast-flow branch of the APCC. Based on the genomic results we suggest that, although current systems in this area are complex and still poorly constrained by direct observations, movements from the tip of the Peninsula toward the Bellingshausen Sea area are frequent enough to facilitate larval dispersal.

Although there is no direct evidence of the larval type of *A. eightsii*, species in the subclass Protobranchia are known to exhibit either pelagic lecithotrophy or brooding larval development [47]. Based on monthly examinations of reproductive tissues of *A. eightsii*, a pelagic lecithotrophic larval development has been suggested for the species [10], a hypothesis that is supported by our results given the correlation between the local ocean current (APCC) direction and the dispersal patterns observed in *A. eightsii*. The APCC is believed to exhibit strong seasonality, being pronounced during November–June and less so during July–October, in concordance with freshwater input cycles [8]. The peak of reproduction and spawning of *A. eightsii* observed between April and May [10] suggests the

dispersal of the species is probably facilitated by the APCC. The population of Yelcho was supported as a source population and thus, an important source of propagules and genetic diversity for recolonization [15]. Efforts to monitor this area are highly recommended if conservation plans need to be taken, especially in a time of rapid climate change at the WAP [48], where increased temperatures and rapid glaciar retreat may impact ocean circulation—and, consequently, gene flow and metapopulation dynamics—in unpredictable ways.

The role of the ACC on species dispersal has been well documented for several taxa, but much less is known in this regard of the importance of other, more localized (and more nearshore) currents. As far as we are aware, this is the first study providing evidence of the role of the APCC on the dispersal of a marine invertebrate species. Our data also suggest that complex and less studied current dynamics may also be important for transporting larva from the tip of the Antarctic Peninsula towards the Bellingshausen Sea area [46], although further genetic, ecological and oceanographic studies are needed to fully understand these dynamics. Additional studies comparing genetic patterns between species with different larval development strategies will provide further insights about the importance of Southern Ocean currents to the dispersal and distribution of Antarctic benthic organisms and their role on the evolution of Antarctic biodiversity.

# 5. Conclusion

Strong genomic differentiation between the Antarctic and South American lineages of *A. eightsii* supported the role of the ACC as a biogeographic barrier between these continental shelves and confirmed previous findings that suggested the existence of two different species (across the Drake Passage). At the Western Antarctic Peninsula, patterns of directional migration were generally consistent with the directional fluxes of the APCC, suggesting a role of this current in the dispersion of the species. Future studies could increase the number of populations and species to test the generality of our findings and incorporate direct measurements of ocean circulation pathways to further investigate the impact and the complexities of the Southern Ocean currents.

Ethical. This study was approved by the ethical committee of the Universidad Catolica de la Santisima Concepcion, Chile.
Data accessibility. All data used in this study, including the COI sequences and genomic data input files are available at the Dryad Digital Repository: (https://doi.org/10.5061/dryad.0cfxpnvxr) [37]. COI sequences are also available in GenBank through accession numbers MT176643–MT176683.
Authors' contributions. C.P.M.-R. conducted the laboratory work, genomic analyses and drafted the manuscript. D.K.A.B., C.J.S., J.S., A.R.-G., S.A.M., L.C., M.M. and A.B. contributed to the fieldwork sampling, provided inputs for the sampling design and drafted and edited the manuscript (or significant parts of it). All authors gave final approval for publication and agree to be held accountable for the content herein.
Competing interests. We declare we have no competing interests.
Funding. This research was supported by an international collaborative project (ICEBERGs), and the following Chilean grants funded by Agencia Nacional de Investigacion y Desarrollo (ANID): CONICYT-NERC (grant no. PII20150078) and Fondecyt (grant no. 1170598) to A.B. and FONDECYT postdoctorado (grant no. 3180331) to C.P.M.-R.
Acknowledgements. We thank the crew of the RRS *James Clark Ross* for their valuable assistance in the field and to Instituto Antartico Chileno (INACH) for permits and sponsoring the project.

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
