## [Reviewer comments · Royal Society Open Science]

Review History

RSOS-200603.R0 (Original submission)

Review form: Reviewer 1

Is the manuscript scientifically sound in its present form?

Yes

Are the interpretations and conclusions justified by the results?

Yes

Is the language acceptable?

Yes

Do you have any ethical concerns with this paper?

No

Have you any concerns about statistical analyses in this paper?

No

Recommendation?

Accept with minor revision (please list in comments)

Comments to the Author(s)

The manuscript intitle "Gene flow in the Antarctic bivalve *Aequiyoldia eightsii* (Jay, 1839) suggests a role for the Antarctic Peninsula Coastal Current in larval dispersal", which was previously submitted to Biology Letters and reviewed by me, incorporated most of the suggestions, making changes in all sections. The writing is clearer, the methodology more detailed, new analyses were included, and the discussion became more robust. As previously mentioned, the manuscript is interesting, technically sound and conducted in a region that is difficult to access and relatively poor explored in phylogeographic and populational genetic terms. Thus, I consider that the manuscript can be accepted for publication.

Minor corrections/suggestions:

Line 109 – Change "*A. eightsii*" to "*Aequiyoldia eightsii*".

Line 122 – Change "and 72° C for 7 min" to "and a final extension of 72° C for 7 min".

Line 149 – There is an empty parentheses ().

Line 153 – Table 2 was not found. Please check.

Line 240 and 241 – Perhaps the expressions "unsurprisingly" and "more surprising" can be replaced by "as expected" and "not expected".

Review form: Reviewer 2

Is the manuscript scientifically sound in its present form?

Yes

Are the interpretations and conclusions justified by the results?

No

Is the language acceptable?

Yes

Do you have any ethical concerns with this paper?

No

Have you any concerns about statistical analyses in this paper?

Yes

Recommendation?

Major revision is needed (please make suggestions in comments)

Comments to the Author(s)

I have revised the second version of this manuscript, and I commend the authors on their efforts to improve it. However, I am still uneasy with the level of missing data being allowed in the population structure and migration analyses. I see that some analyses with less missing data were included in supplementary material, but they are not referred to in the main body of text, and it is not clear if migration tests were performed using that improved dataset. This will need to be clarified before acceptance. Furthermore, the results presented are somewhat contradictory between analyses, but that is not discussed in detail. Therefore, I am recommending major revisions. All my comments can be found in the attached pdf (Appendix A).

I have two other comments here:

What is sample BB2_G1_3? This code does not match the other samples codes

Please revise the supplementary material file, as there is a table without contents, and some of the figures do not say which dataset was used to generate them.

Decision letter (RSOS-200603.R0)

Dear Mr Muñoz-Ramírez,

The editors assigned to your paper ("Gene flow in the Antarctic bivalve *Aequiyoldia eightsii* suggest a role for the Antarctic Peninsula Coastal Current in larval dispersal") have now received comments from reviewers. We would like you to revise your paper in accordance with the referee and Associate Editor suggestions which can be found below (not including confidential reports to the Editor). Please note this decision does not guarantee eventual acceptance.

Please submit a copy of your revised paper before 03-Jul-2020. Please note that the revision deadline will expire at 00.00am on this date. If we do not hear from you within this time then it will be assumed that the paper has been withdrawn. In exceptional circumstances, extensions may be possible if agreed with the Editorial Office in advance. We do not allow multiple rounds of revision so we urge you to make every effort to fully address all of the comments at this stage. If deemed necessary by the Editors, your manuscript will be sent back to one or more of the original reviewers for assessment. If the original reviewers are not available, we may invite new reviewers.

- Data accessibility

It is a condition of publication that all supporting data are made available either as supplementary information or preferably in a suitable permanent repository. The data

accessibility section should state where the article's supporting data can be accessed. This section should also include details, where possible of where to access other relevant research materials such as statistical tools, protocols, software etc can be accessed. If the data have been deposited in an external repository this section should list the database, accession number and link to the DOI for all data from the article that have been made publicly available. Data sets that have been deposited in an external repository and have a DOI should also be appropriately cited in the manuscript and included in the reference list.

If you wish to submit your supporting data or code to Dryad (<http://datadryad.org/>), or modify your current submission to dryad, please use the following link:
<http://datadryad.org/submit?journalID=RSOS&manu=RSOS-200603>

- **Competing interests**

- **Authors' contributions**

- **Acknowledgements**

- **Funding statement**

on behalf of Dr Michael Tobler (Associate Editor) and Kevin Padian (Subject Editor)
openscience@royalsociety.org

Associate Editor's comments (Dr Michael Tobler):

Comments to the Author:

We have received the feedback from two reviewers, both of which provided a generally positive assessment of the manuscript. Before the manuscript can be accepted, however, the authors need to address some substantial issues with their analyses. In addition, both reviewers provided some minor suggestions that should be addressed before the manuscript is acceptable for publication in RSOS.

Reviewers' Comments to Author:

Reviewer: 1

Comments to the Author(s)

The manuscript intitle "Gene flow in the Antarctic bivalve *Aequiyoldia eightsi* (Jay, 1839) suggests a role for the Antarctic Peninsula Coastal Current in larval dispersal", which was previously submitted to Biology Letters and reviewed by me, incorporated most of the suggestions, making changes in all sections. The writing is clearer, the methodology more detailed, new analyses were included, and the discussion became more robust. As previously mentioned, the manuscript is interesting, technically sound and conducted in a region that is difficult to access and relatively poor explored in phylogeographic and populational genetic terms. Thus, I consider that the manuscript can be accepted for publication.

Minor corrections/suggestions:

Line 109 – Change "*A. eightsi*" to "*Aequiyoldia eightsi*".

Line 122 – Change "and 72° C for 7 min" to "and a final extension of 72° C for 7 min".

Line 149 – There is an empty parentheses ().

Line 153 – Table 2 was not found. Please check.

Line 240 and 241 – Perhaps the expressions "unsurprisingly" and "more surprising" can be replaced by "as expected" and "not expected".

Reviewer: 2

Comments to the Author(s)

I have revised the second version of this manuscript, and I commend the authors on their efforts to improve it. However, I am still uneasy with the level of missing data being allowed in the population structure and migration analyses. I see that some analyses with less missing data were included in supplementary material, but they are not referred to in the main body of text, and it is not clear if migration tests were performed using that improved dataset. This will need to be clarified before acceptance. Furthermore, the results presented are somewhat contradictory between analyses, but that is not discussed in detail. Therefore, I am recommending major revisions. All my comments can be found in the attached pdf (RSOS-200603_Proof_hi_reviewed.pdf).

I have two other comments here:

What is sample BB2_G1_3? This code does not match the other samples codes

Please revise the supplementary material file, as there is a table without contents, and some of the figures do not say which dataset was used to generate them.

Author's Response to Decision Letter for (RSOS-200603.R0)

See Appendix B.

RSOS-200603.R1 (Revision)

Review form: Reviewer 1

Is the manuscript scientifically sound in its present form?

Yes

Are the interpretations and conclusions justified by the results?

Yes

Is the language acceptable?

Yes

Do you have any ethical concerns with this paper?

No

Have you any concerns about statistical analyses in this paper?

No

Recommendation?

Accept as is

Comments to the Author(s)

After this new round of review, all my questions were resolved, so the article can be accepted without further corrections.

Review form: Reviewer 2

Is the manuscript scientifically sound in its present form?

Yes

Are the interpretations and conclusions justified by the results?

Yes

Is the language acceptable?

Yes

Do you have any ethical concerns with this paper?

No

Have you any concerns about statistical analyses in this paper?

No

Recommendation?

Accept with minor revision (please list in comments)

Comments to the Author(s)

I commend the authors for a good review and I think the manuscript is ready for acceptance, pending some (very) minor changes for typos and consistency.

Minor changes:

- page 7 line 17: change degrees for the scientific notation (oC)

- page 7 line 62: change structure for STRUCTURE
- revise notation of numbers and decimal points throughout to ensure consistency (some times it has two decimal points, some times three; thousands are some times separated by commas, others not)
- page 8 lines 57-59: I'm not sure what loci the authors are referring to in this section.
- page 9: cite figure 2 when mentioning STRUCTURE results
- page 10 lines 10-18: no reference to mtDNA FST results (and how they do not really match nuclear SNPs FST results)
- Figures: consider be consistent with colours for sampling sites. E.g. O'Higgins is represented in yellow in Fig1 but in green in Fig 3 (Yelcho is yellow in this figure)

Decision letter (RSOS-200603.R1)

Dear Mr Muñoz-Ramírez

On behalf of the Editors, we are pleased to inform you that your Manuscript RSOS-200603.R1 "Gene flow in the Antarctic bivalve *Aequiyoldia eightsi* suggest a role for the Antarctic Peninsula Coastal Current in larval dispersal" has been accepted for publication in Royal Society Open Science subject to minor revision in accordance with the referees' reports. Please find the referees' comments along with any feedback from the Editors below my signature.

Please submit your revised manuscript and required files (see below) no later than 7 days from today's (ie 20-Aug-2020) date. Note: the ScholarOne system will 'lock' if submission of the revision is attempted 7 or more days after the deadline. If you do not think you will be able to meet this deadline please contact the editorial office immediately.

Royal Society Open Science
openscience@royalsociety.org
on behalf of Dr Michael Tobler (Associate Editor) and Kevin Padian (Subject Editor)
openscience@royalsociety.org

Reviewer comments to Author:

Reviewer: 1

Comments to the Author(s)

After this new round of review, all my questions were resolved, so the article can be accepted without further corrections.

Reviewer: 2

Comments to the Author(s)

I commend the authors for a good review and I think the manuscript is ready for acceptance, pending some (very) minor changes for typos and consistency.

Minor changes:

- page 7 line 17: change degrees for the scientific notation (oC)
- page 7 line 62: change structure for STRUCTURE
- revise notation of numbers and decimal points throughout to ensure consistency (some times it has two decimal points, some times three; thousands are some times separated by commas, others not)
- page 8 lines 57-59: I'm not sure what loci the authors are referring to in this section.
- page 9: cite figure 2 when mentioning STRUCTURE results
- page 10 lines 10-18: no reference to mtDNA FST results (and how they do not really match nuclear SNPs FST results)
- Figures: consider be consistent with colours for sampling sites. E.g. O'Higgins is represented in yellow in Fig1 but in green in Fig 3 (Yelcho is yellow in this figure)

===PREPARING YOUR MANUSCRIPT===

If you have been asked to revise the written English in your submission as a condition of publication, you must do so, and you are expected to provide evidence that you have received language editing support. The journal would prefer that you use a professional language editing service and provide a certificate of editing, but a signed letter from a colleague who is a native speaker of English is acceptable. Note the journal has arranged a number of discounts for authors

using professional language editing services
(<https://royalsociety.org/journals/authors/benefits/language-editing/>).

===PREPARING YOUR REVISION IN SCHOLARONE===

-- If you have uploaded ESM files, please ensure you follow the guidance at <https://royalsociety.org/journals/authors/author-guidelines/#supplementary-material> to include a suitable title and informative caption. An example of appropriate titling and captioning may be found at https://figshare.com/articles/Table_S2_from_Is_there_a_trade-off_between_peak_performance_and_performance_breadth_across_temperatures_for_aerobic_sc_ope_in_teleost_fishes_/3843624.

Author's Response to Decision Letter for (RSOS-200603.R1)

See Appendix C.

Decision letter (RSOS-200603.R2)

Dear Mr Muñoz-Ramírez,

It is a pleasure to accept your manuscript entitled "Gene flow in the Antarctic bivalve *Aequiyoldia eightsii* suggest a role for the Antarctic Peninsula Coastal Current in larval dispersal" in its current form for publication in Royal Society Open Science.

on behalf of Dr Michael Tobler (Associate Editor) and Kevin Padian (Subject Editor)
openscience@royalsociety.org

Associate Editor Comments to Author (Dr Michael Tobler):

Reviewer comments to Author:

Appendix A**ROYAL SOCIETY
OPEN SCIENCE****Gene flow in the Antarctic bivalve *Aequiyoldia eightsii*
suggest a role for the Antarctic Peninsula Coastal Current in
larval dispersal**

Journal:	Royal Society Open Science
Manuscript ID	RSOS-200603
Article Type:	Research
Date Submitted by the Author:	10-Apr-2020
Complete List of Authors:	Muñoz-Ramírez, Carlos; Universidad Metropolitana de Ciencias de la Educacion, Instituto de Entomologia Barnes, David; British Antarctic Survey, Biological Sciences CÁRDENAS, LEYLA; Universidad Austral de Chile, Instituto de Ciencias Ambientales y Evolutivas Meredith, Michael; British Antarctic Survey, Morley, Simon; British Antarctic survey, Biosciences Roman-Gonzalez, Alejandro; University of Exeter, College of Life and Environmental Sciences Sands, Chester; British Antarctic Survey (Natural Environment Research Council), Biological Sciences Division Scourse, James; University of Exeter Brante, Antonio; Universidad Catolica de la Santisima Concepcion, Facultad de Ciencias
Subject:	ecology < BIOLOGY, evolution < BIOLOGY, biogeography < CROSS-DISCIPLINARY SCIENCES
Keywords:	Asymmetrical migration, Antarctic Peninsula Coastal Current, NextRAD, Antarctic Circumpolar Current
Subject Category:	Genetics and genomics

Author-supplied statements

Relevant information will appear here if provided.

Ethics

Does your article include research that required ethical approval or permits?:

This article does not present research with ethical considerations

Statement (if applicable):

CUST_IF_YES_ETHICS :No data available.

Data

It is a condition of publication that data, code and materials supporting your paper are made publicly available. Does your paper present new data?:

Yes

Statement (if applicable):

All data used in this study, including the COI sequences and genomic data input files are available at Dryad (https://datadryad.org/stash/share/u30ZZTnVWdldwbDBChsJ1zuni0TVuwRFPVwcv_bQwO8) (Munoz-Ramirez 2019).

COI sequences are also available in GenBank through accession numbers MT176643-MT176683.

Description for each data file is available in the ESM.

Conflict of interest

I/We declare we have no competing interests

Statement (if applicable):

CUST_STATE_CONFLICT :No data available.

Authors' contributions

This paper has multiple authors and our individual contributions were as below

Statement (if applicable):

CPM-R conducted the lab work, genomic analyses and drafted the manuscript. DKAB, CJS, JS, AR-G, SAM, LC, MM, and AB contributed to the fieldwork sampling, provided inputs for the sampling design and drafted and edited the manuscript (or significant parts of it). All authors gave final approval for publication and agree to be held accountable for the content herein.

Gene flow in the Antarctic bivalve *Aequiyoldia eightsii* (Jay, 1839) suggests a role for the Antarctic Peninsula Coastal Current in larval dispersal

Muñoz-Ramírez Carlos^{1,2,3}, Barnes David KA⁴, Cardenas Leyla⁵, Meredith Michael P⁴, Morley Simon A⁴, Roman-Gonzalez Alejandro⁶, Sands Chester J⁴, Scourse James⁶, Brante Antonio^{2,3}

¹Instituto de Entomología, Facultad de Ciencias Básicas, Universidad Metropolitana de Ciencias de la Educación, Santiago, Chile.

²Facultad de Ciencias, Universidad Católica de la Santísima Concepción, Concepción, Chile

³Centro de Investigación en Biodiversidad y Ambientes Sustentables (CIBAS), Universidad Católica de la Santísima Concepción, Concepción, Chile

⁴British Antarctic Survey, Natural Environment Research Council, Cambridge, UK

⁵Centro Fondap IDEAL, Instituto de Ciencias Ambientales y Evolutivas, Facultad de Ciencias, Universidad Austral de Chile, Valdivia, Chile

⁶College of Life and Environmental Sciences, University of Exeter, Penryn, Cornwall, TR10 9EZ, UK

Keywords: Assymetrical migration, Antarctic Peninsula Coastal Current, NextRAD

*Author for correspondence (abrante@uesc.cl).

†Present address: Department, Institution, Address, City, Code, Country

1. Summary

[revised manuscript text omitted]

Acknowledgments

We thank the crew of the RRS James Clark Ross for their valuable assistance in the field and to INACH for granting the necessary permits. This study was funded by a FONDECYT Postdoctorado grant (3180331) and a CONICYT-NERC Collaborative Project, ICEBERGS (PII20150078).

Ethical Statement

This study was approved by the ethics committee from Universidad Catolica de la Santisima Concepcion, Concepcion, Chile.

Funding Statement

This research was supported by an international collaborative project (ICEBERGS), a CONICYT-NERC grant (223449) to AB and a FONDECYT postdoctorado grant (3180331) to CPM-R.

Data Accessibility

All data used in this study, including the COI sequences and genomic data input files are available at Dryad (https://datadryad.org/stash/share/u30ZZTnVWldwdbDBChsJlZuni0TVuvRFPVwcv_bQwO8) (Munoz-Ramirez 2019). COI sequences are also available in GenBank through accession numbers MT176643-MT176683.

Competing Interests

We have no competing interests

Authors' Contributions

CPM-R conducted the lab work, genomic analyses and drafted the manuscript. DKAB, CJS, JS, AR-G, SAM, LC, MM, and AB contributed to the fieldwork sampling, provided inputs for the sampling design and drafted and edited the manuscript (or significant parts of it). All authors gave final approval for publication and agree to be held accountable for the content herein.

References

[revised manuscript text omitted]

Table 1: Sampling localities for *Aequiyoldia eightsi*. N= Number of individuals sampled. Last two columns show the number of loci obtained from the NextRAD protocol when a minimum of four (min4) or ten (min10) individuals having the locus were set as the criteria to include the Locus in the data set. Loci count are average per population. N= Number of samples, SA= South America, WAP= Western Antarctic Peninsula.

Site	Region	Latitude	Longitude	N	M4	M10	M20
Magellan Strait	SA	-52.4231	-69.5717	8	87067	35046	729
Puerto Williams	SA	-54.9258	-67.9911	6	39621	16944	558
O'Higgins Base	WAP	-63.3208	-57.8986	8	39615	16753	440
Yelcho	WAP	-64.8758	-63.6089	10	22077	10146	224
Rothera	WAP	-67.5709	-68.1269	9	104922	42349	746

***R. Soc. open sci.* article template**

Figure 1: A, study area and sampling localities for the species *Aequiyoldia eightsii* including an haplotype network inferred from the mtDNA (COI) data set and a simplified scheme of the ocean currents present in the Western Antarctic Peninsula (ACC, Antarctic Circumpolar Current; APCC, Antarctic Peninsula Coastal Current; CC, Antarctic Coastal Current (from Moffat & Meredith 2018)). Haplotype colors are matched with their corresponding geographical distribution. ACC, Antarctic Circumpolar Current; APCC, Antarctic Peninsula Coastal Current; CC, Antarctic Coastal Current (Moffat & Meredith 2018). B) Geographical distribution of the genus *Aequiyoldia* (Linse, 2014).

Figure 2: A, cluster analyses (up) and maximum likelihood phylogenetic tree (down) for *Aequiyoldia eightsi* populations from Antarctica and South America using the NextRAD dataset. Top cluster plot represents independent analysis within regions. B, Depiction of asymmetric gene flow between populations from the Western Antarctic Peninsula calculated with divMigrate-online using the D statistic of Jost (2008) and 999 bootstrap permutations. Wider and bolder arrows represent stronger gene flow in the direction of the arrows. Graphics modified from original (figure S1, ESM) for illustrative purpose.

Figure 3: Discriminant analyses of principal components, DAPC, for individuals of the bivalve *Aequiyoldia*
*eightsii* from South America and Antarctica based on single-nucleotide polymorphism data. A, Discriminant
Analysis plot. B-C, DAPC membership plots for $k=5$ (B) and $k=2$ (C) cluster assignments. DA= Discriminant
analysis. Samples with the lowest number of loci are labeled in the DAPC plot (A).

Dear Chief Editor,

We are glad to having this manuscript transferred to the RSOS journal. Here we present a point-by-
point response to all the reviewers questions and suggestions from the previous review process in
Biology Letters. We have uploaded a version with all changes tracked under the “Review File”
designation so the editor can check we have made all the required changes. A clean version
incorporating all these changes is provided as the main manuscript using the journal template.

Response to reviewers’ comments and questions.

Editorial Board comments:

Your paper has received some mixed responses from reviewers. In summary, some greater explanation
of the rationale for the data collected and some reanalysis and expanded analysis of those data would
be needed before the study is ready for publication. If you can do this (and refine the writing) I think
that this work could be valuable, especially to an audience of marine ecologists. On the other hand, my
feeling is that if you were to implement these revisions, you would really struggle to fit the manuscript
into the short Biology Letters format. So in my opinion the paper would be better off in a marine or
molecular ecology journal that would allow the work to be properly developed and explained. There is
also the option to transfer the submission to Royal Society Open Science, if you are able to address the
reviewer's comments in a revision.

**Thank you and all the reviewers for the comments and suggestions, which definitely improved**
**our manuscript.**

**Several common suggestions arose, to which we have provided careful consideration. For**
**example, all reviewers pointed to expand discussion to account for a puzzling pattern of**
**migration from O’Higgins to Rothera. We have done so and included more literature and an**
**expert on ocean circulation was invited to contribute on this. Also reviewers pointed out to**
**adding more methodological details and additional analyses. We have now, with no space**
**restrictions, added substantial methodological details and also added a DAPC analysis to contrast**
**our previous structure results.**

**Although not always was possible to incorporate all suggestions because sometimes they were**
**contradictory, all comments were carefully considered one way or another. For example, when**
**the suggestion of one reviewer was taken, the others were taken as a call for us to better clarify or**
**explain the problem at question.**

Reviewer Comments to Author:

Referee: 1

Comments to the Author(s)

This paper seeks to understand gene flow and the role of the Antarctic Peninsula Coastal Current in
larval dispersal. This is an interesting region that is not well studied.

In my opinion the manuscript would benefit from 1) additional analyses, 2) more comprehensive
analyses, 3) careful proof reading.

The conclusion, last line of the abstract and title are too strong given the analyses. The analyses need to
be more comprehensive in order to reach this conclusion. The paper would benefit from an additional,
alternative method(s) to explore this result and to determine if the same result is still obtained. Options
include GeneClass2 or BayesAss for examples.

**Response: We realized that GeneClass2 is very sensitive to missing data and it does not perform**
**well if data set do not have high completeness. Because our data have missing data, we tried**
**BayesAss instead. However, this program did not perform well to test asymmetrical migration.**
**After attempting several runs, results produced very similar migration rates among most**
**populations, perhaps because the internal models assume homogeneous migration rates or at**
**least they are more highly weighted. In these models, a large number of parameters are**
**estimated simultaneously involving complex optimization algorithms, resulting in intensive**
**computational requirements, which may have complicated the reaching of parameter**
**convergence. As a consequence, these models are often used as black boxes implying that users,**
**due to the complexity of these analytical approaches, typically only have a limited understanding**
**of the underlying models and their assumptions. Thus, we are not in a position to adequately**
**assess the application of these methods is inappropriate. Furthermore, it is possible that these**
**analysis pick asymmetric signals only when these are very strong and not subtle. However, we**
**have included this analysis (BayesASS) to show what we found, but, due to the low information**
**gained with it, we still focus on the divMigrate results for our discussions.**

Even with additional analyses concluding that the APCC is the driver is too strong - particularly
without an expanded Discussion. The introduction commences with 'The evolution of Antarctic
biodiversity has been largely influenced by ocean currents.'

**Response: Our conclusion is that the data suggest a role for the APCC and offers a workable**
**hypothesis to be tested in future studies. That is not necessarily a strong claim in our opinion, but**
**the expanded discussion is now provided.**

There is data missing that is available on GenBank that should be included in the COI part of their
study. This will help to give a more comprehensive overview of the phylogeographic patterns.

**Response: There is only one study which data (COI data) would expand our analysis (Gonzalez-**
**Wevar et al. 2019). Unfortunately, the data was not made available in GenBank, neither the**
**author of the paper was able to locate the data when asked.**

The methods (from library prep to bioinformatics and data analyses) need to be explained much further
because there are lots of gaps at the moment (MAF, HWE etc?). How were the different number of loci
46 per site obtained?

**Response: We have conducted several additional analyses that expanded our results and they are**
**presented now in the manuscript. The reductionist approach was previously kept only to fit the**
**length constraint of the journal and to that extent, we used only those analyses that we thought**
**were more direct and less redundant. Nevertheless, we have added now much more detail**
**regarding the processing of the genomic data, criteria for filtering and background that support**
**some of the decisions taken in regards of analyses chosen.**

I cannot see Tables S3 and S4.

**Response: Probably an incompatibility between OS. I made them with Linux's libreOffice. We**
**have moved the text and tables to the template of the RSOS, so we hope this is fixed.**

The introduction (and discussion) talks of the importance of oceanic currents on influencing the
evolution of Antarctica. However, something that is overlooked is glaciation. Glacial cycles (on annual
and longer time scales) would have a massive influence on the habitat on this region. What impact
might they have on the phylogeographic patterns that are seen?

**Response: Good question! It seems....past history can influence the patterns, so results should be**
**taken with caution. It is difficult to know exactly how phylogeographic history might impact.**
**However, some of the issues can be attenuated by avoiding analysis on divergent lineages and**
**structured populations. There was evidence from previous studies that cryptic lineages were**
**present in this area, so the COI marker allowed us to confirm we were working with a single**
**lineage in the WAP. The analysis also has the advantage of working on statistics that are not**
**severely biased by genetic diversity (e.g. D). This alleviates, at least in part, a strong influence of**
**past bottlenecks or demographic expansions.**

- an oceanographic modelling component might help support the conclusions - see Oceanography
promotes self-recruitment in a planktonic larval disperser PR Teske, J Sandoval-Castillo, E van Sebille,
J Waters, LB Beheregaray. Scientific Reports 6, 34205

**Response: This is a great tool that, unfortunately, cannot be implemented straightforwardly in**
**our study due to the low number of sites. Downstream analyses that depend on the results of**
**these simulation approaches take pairwise comparisons between sites, which in our case would**
**produce too few data to perform statistical testing. We believe that the analyses we have**
**implemented are in accordance with the data available and provide enough evidence for our**
**conclusions. However, we have expanded our discussion to include ocean circulation evidence**
**that may support our results and provide a plausible explanation for the single pattern that may**
**had deviated from the general expectation (Migration from Ohiggins to Rothera).**

Minor comments:

- the species name is spelt incorrectly throughout the paper

**Resp: Fixed.**

- the title should be 'suggests' but I don't that said I think the title is too bold given the analyses and the
results.

**Resp: The Title has been reworded. More analyses and results were performed, some of which**
**support previous analyses, although others were non-informative. The word “suggest” is kept for**
**caution.**

- what is the total distribution of the species? Show this on the map.

**Resp: Added to the map in figure 1.**

- how deep does this species occur at? I think there are records down to 3000 m. (does the species
occur between the tip of South America and Antarctica?)

**Resp: *Aequiyoldia eightsii* normally lives in the first 100 meters depth, although there are rare**
**records of specimens collected as deep as 800. There are no records of the species between the tip**

**of Southern South America and Antarctica. The northern record of the Antarctic group is Signy**
**Island, which is not far from the tip of the Antarctic Peninsula.**

- line 80 - why will this help understand population size?

**Resp: We have changed the word to Effective population size, a helpful parameter in population**
**genetics and conservation genetics. This parameter is closely related to genetic diversity so by**
**knowing this parameter, inferred from genetic diversity, can inform the health of the population**
**in genetic terms.**

- double check your use of North and south in this paper and maps

**Resp: Most text regarding currents and positions was re-organized and reworded.**

- line 88 - the table does not show 42 but rather 41.

**Resp: Changed. It was a typo.**

- what are the short comings of the divMigrate method?

**Resp: The method has shown good efficiency and accuracy in testing asymmetrical migration**
**rates based on empirical and simulation studies. It is only implemented for testing migration and**
**therefore, it does not provide estimations for other genetic summary statistics.**

- what do the numbers in Figure 2B represent?

**Resp: They represent migration rates in the direction of the arrows.**

- line 155 - try DAPC - it might work better

**Resp: DAPC was added now and it was consistent with STRUCTURE.**

- line 181 Where do the 7,781 SNPS come from?

**Resp: More info was added regarding this. These number of SNPs were the result of filtering loci**
**by the number of individuals that carried them. In this specific case, 7,781 were the number of**
**loci that were present for 10 or more individuals.**

- there are many spelling mistakes throughout. One of the native English speakers should proof read
carefully.

**Resp: It was now carefully checked by all co-authors.**

- did you test for an evolutionary model before building the tree? Please report this.

**Resp: The use of an evolutionary model of our choice is not implemented in the RaxML program**
**(Stamatakis 2008). It only contains the GTR+Gamma model. The argument is that the model is**
**complex enough to fit the heterogeneous nature of a data set the size of a genome (or a random**
**subset of it), and therefore, suitable enough for the large data sets for which it was created. This**
**aspect was now informed in the method section.**

- the APCC arrow in Figure 1 looks to be going in both directions? How does this support your case?

**Resp: The APCC in the area described flows southward between Yelcho and Rothera and**
**northward between Yelcho and Ohiggins. This circulation pattern matches the migration pattern**

**found between these populations. But it is true that this cannot explain the southward migration**
**between Ohiggins and Rothera. We have now provided an expanded discussion to explain this**
**pattern, providing a hypothesis regarding a southward flow that goes some kilometers away**
**from the coast and it is time dependent. This explanation is grounded in oceanographic evidence**
**and provided by an expert in Antarctic currents, Dr. Mike Meredith (BAS), who was invited to**
**contribute with his expertise to this work as co-author.**

Referee: 2

Comments to the Author(s)

The manuscript "Asymmetric gene flow in the Antarctic bivalve *Aequiyoldia eightsi* suggest a role for
the Antarctic Peninsula Coastal Current in larval dispersal" aims to assess and understand the role of
local oceanographic currents in an Antarctic bivalve in the Western Antarctic Peninsula.

The manuscript is well written, the sampling design is adequate for the questions asked and the markers
employed sufficient. Regarding data availability, although the input files were made available in
DRYAD, the code (as requested by the journal) was not. Overall, I think the manuscript is of limited
interest to the broad readership of Biology Letters, and would benefit from a longer format in a more
specialised journal such as Journal of Biogeography or Molecular Phylogenetics and Evolution.

In particular, authors should develop the rationale behind the different datasets (in terms of number of
loci) used in the different analyses. In total, this study has 41 samples, but in some cases SNPs were
retained if they were present in only 4 individuals (10% of total dataset) or 10 individuals (24% of total
dataset). These are quite low values, in a field that strives to keep up to 80% of all individuals. This
would suggest that the RAXML analyses were performed with a very high level of missing data.

**Resp: There is an undeniable compromise between number of loci an amount of missing data.**
**Recent studies have shown that is highly desirable to prefer more loci over missing data (Huang**
**& Knowles 2016) because loci with high levels of missing data are representative of the high**
**mutation-rate spectrum allowing the resolution of shallow relationships. Indeed, reducing**
**missing data can introduce biases. So this was the main reason to use this data set in our analysis.**

Furthermore, the methods detailed in Sup Material suggest that Structure and migration analyses were
based on SNPs found in 7 individuals (17% of total dataset), which is not mentioned in the methods or
result sections of the main document. The authors should consider redoing the analyses to include
SNPs that have a higher call rate.

**Response: 7 individuals is the minimum. The data set therefore contains loci with a range of**
**coverage from 7 to 41 individuals presenting a given locus. Some analyses are sensible to missing**
**data, others are not. Still, the behavior of some in regards of missing data is not well known.**
**Here, we have used analyses that perform relatively well with high levels of missing data. Indeed,**
**in some of our analysis, this level of missing data is even desirable to avoid biases. That is why we**
**use different levels of missing data in different analysis, to meet some minimum requirements of**
**some of these analyses.**

**Anyway, these information and methodological details are now provided in the methodology**
**section.**

Finally, the section relating observed genetic structure with local current patterns and life history

features needs to be expanded. According to Figure 1, O'Higgins Base appears to be somewhat in the
confluence of two local currents, and the APCC is flowing S-N in this region (as opposed to N-S in the
remainder of the Peninsula), how does that reconcile with the inferred N-S migration linking O'Higgins
Base to Rothera?

**Resp: The discussion has been expanded to treat this aspect with more detail. Including literature**
**that was not included in the previous version of the manuscript that suggest potential**
**mechanisms that can explain the migration patterns. We have also reword some of the ideas to**
**state some of our statements as tentative explanations or hypothesis that could be tested in future**
**studies.**

And could that be linked with the choice of parameters used to sub-setting the data?

**Response: This is a great suggestion. We have redone the analyses varying the level of missing**
**data. Results show no dependency to the filtering processes in this regard. These changes were**
**incorporated now to the corresponding sections of the main text.**

As such, I'm recommending rejection, and suggest a resubmission to another journal.

I have added other comments in the attached document that I hope will be useful to the authors going
forward.

Referee: 3

Comments to the Author(s)

The manuscript (RSBL-2019-0883) entitled "Asymmetric gene flow in the Antarctic bivalve
*Aequiyoldia eighitsi* suggest a role for the Antarctic Peninsula Coastal Current in larval dispersal"
conducts a genetic survey in southern South America and in Antarctic peninsula to determine the
patterns of population genetic structure and gene flow among five localities. The main justification is to
evaluate the contribution of Antarctic Circumpolar Current (ACC) and Antarctic Peninsula Coastal
Current (APCC) in isolating or connecting populations of marine benthic organisms, respectively. To
assess this, they use two molecular approaches: (1) COI gene (mtDNA) sequencing; and (2) genotyping
of single nucleotide polymorphism (SNP) through nextRAD. The authors concluded that ACC acts
isolating the South America and Antarctica *A. eighitsi* populations, while the overall gene flow among
Antarctic peninsula localities follow the major direction of APCC. The article is interesting, technically
sound and conducted in a region that is difficult to access and relatively poor explored in
phylogeographic and populational genetic terms. Therefore, its publication, after corrections, seems
appropriate to me.

Comments:

1) I recommend doing a careful English review. I identified a few minor errors. Since English is not my
native language other grammatical and spelling errors may have remained unnoticed.

**Response: A careful review have been conducted by one of our English speaker co-authors.**

2) Mainly in the Introduction section the English should be improved to facilitate comprehension and
improve readability.

**Response: Done.**

3) Title. Just a suggestion: “Gene flow of Antarctic bivalve *Aequiyoldia eighitsi* suggests that larval
dispersion is mediated by Antarctic Peninsula Coastal Current”.

**Response: The proposed title seems appropriate, so we accept the change from Asymmetric gene**
**flow to gene flow. However, given that the sampling covers a portion of the distribution and not**
**the entire distribution of the species, we prefer to keep part of the title in the original form to**
**emphasize the potential role of the local current in the local area, but not ruling out the role for**
**other currents somewhere else in the distribution of the species. In other words, it may be**
**possible that the ACC still play a role elsewhere, but we have only data to support the role of the**
**APCC (vs the ACC) in the study area.**

4) Line 44. Would it be possible to provide a reference for the first sentence?

**Resp: Barker et al. 2007 and Griffiths 2010 are excellent sources to cite in this regard. They have**
**both been added to the manuscript. (CM-R)**

**Barker PF, Filippelli GM, Florindo F, Martin EE, Scher HD. 2007. Onset and role of the Antarctic**
**Circumpolar Current. Deep Sea Res II. 2007;54:2388–98. doi: 10.1016/j.dsr2.2007.07.028.**

**Griffiths HJ (2010) Antarctic marine biodiversity - what do we know about the distribution of**
**life in the southern ocean? PLoS One 5: e11683. doi: 10.1371/journal.pone.0011683**

5) Line 44 to 56. This paragraph should be reformulated. First you say that ACC isolate Antarctica
marine fauna and after you say that ACC has an important role in promote dispersal. I think you need
specify that ACC promotes the dispersal of marine organisms around the Antarctic and the isolation in
relation to other continents than Antarctic. The first phrase of summary explains this, however here the
information got a bit confusing.

**Resp: We have reworded these sentences and added more text to make it more clear. Thanks for**
**the suggestion.**

6) Line 58. The sentence, “...follows a complex pattern among the many islands of the WAP.”, should
be reformulate to something like “a more complex circulation patterns along the coast, ...”, as observed
in abstract. I made a suggestion in the attached document.

**Resp: We have reworded now as suggested, with a slight modification to add that the circulation**
**pattern is mainly due to the physical oceanographic characteristics rather than the shape or**
**presence of the islands.**

7) Line 75. Considering starting a paragraph in “Genome-wide molecular data...”.

**Resp: This text has been moved to start a new paragraph as suggested and new information and**
**additional details were provided.**

8) Line 80. “...population size,...” change to “... effective population size,...” Keep in mind the
difference between population size and effective population size. This is important because of the

difference of what is a population in ecological and genetic/evolutionary terms. This issue leads to the
difference in demographic connectivity and genetic connectivity. That is, detecting that there is genetic
connectivity does not necessarily mean that there is demographic connectivity. Thus, the desired
contribution of gene flow to the resilience of populations may not be realized. Two interesting texts on
this issue are Cowen & Sponaugle 2009 (which you already mention) and Lowe & Allendorf 2010
("What can genetics tell us about population connectivity?" *Molecular Ecology*, 19, 3038-3051).

**Resp: Fixed. Thank you for pointing this out.**

9) Line 80. "... adaptation,...". How can a study of gene flow contribute to a better understanding of
the adaptation process? I think the importance of studying gene flow in understanding adaptive
processes is quite indirect.

**Resp: True, the relationship is not as direct as in the other processes, but it is still important and**
**worth mention it. High levels of gene flow act as an homogenizing force that will prevent local**
**adaptation. On the other hand, maladaptive genes can easily spread through the population if**
**gene flow is high. Also, well connected populations contribute may contribute to an overall larger**
**effective population size, and as such, reducing genetic drift and increasing the relative**
**importance of selection forces, if there is any. So, although less a direct force it is an important**
**parameter that can influence how adaptations may establish and spread.**

10) Line 80-81. "... and resilience,... warming Antarctica.". Perhaps it is better to explain a little more.
It can be mentioned that determining the direction of gene flow can contribute to discovering source
and sink populations and this can play an important role in maintaining populations, especially
considering climate change.

**Resp: Added explanations and some citations to support the explanations.**

11) At the end of the Introduction you comment about conservation issues, like global warming, so you
could use conservation terms like source and sink populations in your discussion and conclusion. For
example: you can conclude that RO is a sink population while YEL is a source population. In this way,
is expected a higher genetic diversity in BO than in YEL and BO.

**Resp: Thanks for the suggestion, we have included an extended discussion and we have**
**introduced these terms accordingly.**

12) Line 88. According to table 1 there are forty-one individuals.

**Resp: Yes, there were 42 individuals sequenced originally, but one specimen yielded almost no**
**data (extremely low quality) so it was of no use to include it in the final data set. We have fixed**
**this discrepancy now in the methodology section. It was a typo from a previous draft.**

13) Line 90. Throughout the text it is better to cite COI than mtDNA (or use COI mtDNA). Using
mtDNA can confuse the reader and lead to the misconception that the entire mitochondrial genome has
been used. The term "COI" for the cytochrome c oxidase subunit 1 mitochondrial gene is widely used
and recognized.

**Resp: Fixed. mtDNA word was replaced by COI through the entire text.**

14) Line 90. Have the COI sequences deposited in any public database (e.g. GenBank)?

**Resp: Yes, they are now, in addition to the Dryad submission. Acc. numbers MT176643-**
**MT176683 were added to the Methodology and Data sections.**

15) Line 104. I think this can be written in a better way. Example: “The SNPs generated by nextRAD
was then used...”. The molecular marker that you use is SNPs, the methodology that you use to obtain
it was the nextRAD.

**Resp: Changed in all the instances. Thank you for the suggestion.**

16) Line 107. I have some questions about these numbers. First, since this species does not have a
reference genome, the number of loci obtained in nextRAD methodology need to be estimated
arbitrary. What criteria were used to define the number of loci? Second, the number of sites (5,973,095)
corresponds to the total number of bases pairs summing all loci, but you could also provide basic
statistics regarding loci size (mean, minimal/maximal loci size and standard deviation). If the number
of words is at the limit it can added in supplementary material. Third, it is important to make it clear
that this number is the number of bp and not necessarily the number of SNPs. So, I suggest informing
the proportion of these sites that are polymorphic.

**Resp: The reviewer is right and due to the limitation in the number of words many of these**
**methodological details were initially described in the Supplementary, while others were**
**assumed as known. Now, with less word-count limitations, we have moved several methodological**
**details to the main text and have also added additional explanations to previously less clear**
**aspects of the methods.**

**The criteria was based on what the different analyses allowed for missing data. In general, a**
**greater amount of missing data also means a greater amount of loci. However, because some**
**analyses cannot perform well with too much missing data (e.g. PCA), we increased the amount of**
**coverage (minimum % of individuals with data to include a loci; below that % the locus is**
**filtered out) for those analyses at the cost of the amount of loci retained. On the other hand, other**
**analyses are less sensitive to missing data, or even, it is desirable to include missing data because**
**missing data can still be informative for some samples and it because ensures a longer alignment.**
**Such is the case of RAxML that performs very well with missing data and it has been shown that**
**including missing data for phylogenetic reconstructions is preferable (Huang & Knowles 2016).**

**Finally, we have followed the suggestion of adding some basic summary statistics of the data to**
**the results section.**

17) Line 112. The final number of SNPs used were 7781. First, what was the initial number of SNPs
before applying the quality control filters? It could be added in the supplementary material. Second,
some authors used only the first SNP of each loci to avoid data dependency. Have you applied any
filters to prevent the same locus from contributing to multiple SNPs, or does the distribution of the
SNPs in the genome itself ensure that each locus will usually only have 1 or 2 SNPs?

**Resp: The total amount of prefiltered loci was 352,263. After several quality filters the number**
**decreased to 297,417. Then, depending on the minimum number of individuals required for a**

**locus to be written out this number was strongly reduced. In the case of setting this value to 10**
**(moderately stringent), mean that each locus in the matrix needed to be represented by at least 10**
**individuals (25 % sample coverage). 8808 loci met this criterium and 7781 were variable at least**
**at one site (one SNP). Some of these loci had more than one SNP, in which case only one SNP was**
**retained (selected randomly). Given that the data is generated as a random reduction of the**
**genome (<<< 1%), the assumption of independence of each loci is realistic. Only when more than**
**one variable site is present at a locus, a selection of only 1 must be made to ensure independence.**
**The iPyRAD algorithm is to chose 1 variable site at random when this is the case for a locus in**
**order to generate a SNP matrix.**

**These details were added now to Methodology and result sections.**

18) Line 119. Another way to explore your data regarding genetic structuring is through PCA. A
relatively simple way to do PCA is through the adegenet package (which you used to calculate the
value of mtDNA). I think this would reinforce its results, since it is a method that uses other criteria to
look at genetic structuring. Moreover, it is fast and simple to run. If the number of figures is at the limit
it can be added in supplementary material.

**Resp: Good suggestion. Now with the extra-space we have added this analysis which also**
**respond to first two reviewers who suggested additional clustering analyses/tests.**

19) Line 132. What is the size of the sequenced fragment? You should include this information in
Material & Methods or in Results section (or at least in the Supplementary Material).

**Resp: The fragment is 629 bp. Added to methodology.**

20) Line 133 and 134. Provide the percentage value of the genetic distance and inform which model
you used to calculate (p-distance, K2P,...). This information is important to facilitate comparison of
your results with other studies.

**Resp: Added percentages to Results and computation details to Methods.**

21) Line 177. Wouldn't it be in ACC direction favor? If ACC flows south to north (as in Figure 1) and
the brittle star has gene flow from the Antarctic Ocean to South America, then would it be in favor of
ACC, right?

**Resp: Apologize if this was not clearly put in the text, but the ACC current flows circumpolarly**
**around antarctica. So the fact that in the portion of the West Antarctic Penninsula flows in a**
**South-West to North-East direction is only locally. These sentences were now reworded to clarify.**

22) Line 194 to 197. Halfway from BO to RO the gene flow goes against the direction of the APCC, as
from YE the APCC flows towards to the Bransfield Strait. So, in part, there is an inconsistency
between the current direction and the results. I wonder what happens after APCC meets CC? I do not
know the behavior of currents in this region, but can the CC have any characteristics that help explain
the high gene flow from BO to RO, even considering that up to halfway from BO to RO the current
flows in the opposite direction of gene flow?

**Resp: This is an excellent question that was not easy for us to explain. We realized that**
**circulation models in Antarctica are still in its infancy and there is much to learn about their**

**dynamics. This is why we consulted this aspect to a well known expert on Antarctic Ocean**
**circulation, Dr. Mike Meredith, who has worked in the study area and has co-authored papers on**
**these currents. He provided us with potential explanations for this pattern, along with several**
**valuable insights in this regards. We have added these explanations to the discussion section and**
**we have also invited Dr. Meredith to join the MS as a co-author for his contributions.**

23) Another question that could be commented is the different results observed between COI and SNP
data. While COI indicate a higher similarity among RO and YEL, SNPs indicate that BO and RO are
more similar.

**Resp: Yes. It is not surprising to observe these differences for several reasons. The COI gene is**
**much less variable as a marker compared to SNPs and as such, low variation is more prone to**
**provide less resolutive results. Secondly, the coi gene tends to reflect older dynamics, whereas**
**SNPs have a tendency to reflect more recent processes, even at the ecological scale, so the former**
**is prone to historical processes while the later is more prone to reflect current or recent processes**
**(e.g. migration). Finally, the COI gene represent the history of a single gene and it is well known**
**that different genes often have different phylogenetic histories due to incomplete lineage sorting,**
**differential gene migration, etc. This is why analysing multilocus data offers greater resolution**
**and less biased inferences on average.**

24) References. There are some minor format inconsistencies in references. Please, check. Examples:
“Clarke, A., Crame, J., 1989. ...”
“Clarke A, Barnes DKA, Hodgson DA 2005 ...”
“Eaton, D. A. R. 2014. ...”
“Barnes DKA, Peck LS (2008) ...”

**Resp: Thank you. They were now corrected.**

25) Some minor suggestions were made in the attached document.

**Resp: Thanks, they were carefully reviewed and taken into consideration.**

~~G~~Asymmetric gene flow in the Antarctic bivalve *Aequiyoldia eighitsi* suggest a role for the
Antarctic Peninsula Coastal Current in larval dispersal-
Muñoz-Ramírez Carlos^{1,2,3}, ~~Barnes David KA⁴, -Cardenas Leyla⁵, Meredith Mike⁴, Morley~~
~~Simon A⁴, Roman-Gonzalez Alejandro⁶~~, Sands Chester J⁴, ~~Barnes David KA⁴~~, Scourse James⁶⁵,
~~Roman-Gonzalez Alejandro⁵, Morley Simon A⁴, Cardenas Leyla⁶~~, Brante Antonio^{2,3}

¹Instituto de Entomología, Facultad de Ciencias Básicas, Universidad Metropolitana de Ciencias de la
Educación, Santiago, Chile.

²Facultad de Ciencias. Universidad Católica de la Santísima Concepción. Concepción. Chile

³Centro de Investigación en Biodiversidad y Ambientes Sustentables (CIBAS). Universidad Católica de la
Santísima Concepción. Concepción. Chile

⁴British Antarctic Survey, Natural Environment Research Council, Cambridge, UK

~~⁵Centro Fondap IDEAL, Instituto de Ciencias Ambientales y Evolutivas, Facultad de Ciencias,~~
~~Universidad Austral de Chile, Valdivia, Chile~~

~~⁶⁵College of Life and Environmental Sciences, University of Exeter, Penryn, Cornwall, TR10 9EZ, UK~~

~~⁶Centro Fondap IDEAL, Instituto de Ciencias Ambientales y Evolutivas, Facultad de Ciencias, Universidad Austral de Chile, Valdivia, Chile~~

* Corresponding author:

Dr. Antonio Brante

Departamento de Ecología, Facultad de Ciencias, Universidad Católica Ssma. Concepción,

Alonso de Ribera 2850

Concepción, Chile.

Tel: +56 41 2345642; Fax: +56 41 2345251

abrante@ucsc.cl

Abstract

The Antarctic Circumpolar Current (ACC) both isolates and connects the Southern Ocean
biodiversity. However, the role of other Antarctic ocean currents, like the Antarctic Peninsula
Coastal Current (APCC), is relatively unknown in terms of driving biological processes. In the
West Antarctic Peninsula (WAP), the ACC flows in a South~~west~~-North~~east~~ direction, whereas
the APCC follows a more complex circulation pattern along the coast, with ~~flows~~-southwards
~~and~~ northward ~~movements~~ depending on the ~~location~~area. Using genomic data (~~NextRAD~~),
we estimated genetic structure and ~~asymmetric~~ migration ~~rates~~ between populations of a benthic
bivalve from the shallows of southern South America and the West Antarctic Peninsula with the
aim to test the role of the ACC and the APCC in its dispers~~ional~~ ~~potential~~. We found strong
genetic structure across the ACC (between southern South America and Antarctica) and
moderate structure between populations of the West Antarctic Peninsula. ~~Rates of~~
~~asymmetric~~ ~~Migration rates~~ ~~gene flow~~ ~~within the WAP~~, ~~which were strongly biased in a North-~~
~~South direction~~, were highly consistent with a role of the APCC on species dispersal. ~~In addition~~
~~to~~ ~~Along with~~ supporting current knowledge about models of ocean circulation in the West
Antarctic Peninsula, ~~the North-South pattern of gene flow~~ ~~migration from the tip of the Antarctic~~
~~Peninsula from the northernmost population of the Peninsula to the Bellingshausen Sea~~
~~suggests~~ ~~highlights the complexities of Antarctic ocean circulation~~ ~~an extended North-South~~
~~influence of the APCC~~. This study provides novel biological evidence of a role of the APCC as a
driver of species dispersal and highlights the power of genomic data for aiding in the
understanding of complex oceanographic processes.

Introduction

The ~~evolution of~~ Antarctic biodiversity has been largely influenced by oceanic currents (Griffiths
2010; Barker et al. 2007). The formation of the Antarctic Circumpolar Current (ACC) some ~30
Mya, that flows clockwise (~~west to east~~) around Antarctica, has been regarded as one of the main
events driving the isolation of the marine fauna (Clarke and Crame, 1989). The absence of
sharks, rarity of other groups like rays and crabs, and the flourishing of yet others like
pycnogona, polychaete worms and peracarid crustaceans exemplify the isolation and uniqueness
of Antarctic biodiversity (Thatje et al. 2005). Along with serving as an semi-isolating force
(Clarke et al. 2005), the ACC has ~~also~~ been regarded as an effective dispersal agent within the
Southern Ocean, facilitating migration around Antarctica. Large circumpolar ranges and patterns
of genetic connectivity of several taxa including the limpet *Nacella concinna* (Gonzalez-Wevar
2016), the octopus *Pareledone aequipapillae* (Allcock et al. 2011), and the crinoid
*Promachocrinus kerguelensis* (Hemery et al. 2012) are only a few examples supporting the role
of the ACC as a driver of dispersal for marine fauna. Comparatively Little is known, however,
about other Antarctic currents and their potential role on benthic species dispersal.

In contrast to the ACC, the Antarctic Peninsula Coastal Current (APCC), located in the Western
Antarctic Peninsula (WAP), follows a more complex circulation pattern among the coast and the
many islands of the WAP, forced by freshwater discharge and downwelling-favourable winds
near the coast (Moffat & Meredith 2018). From Anvers Island, it moves in opposite directions
along the coast of the WAP. One branch moves in a Northeast (NE) direction toward the tip of
the Antarctic Peninsula, whereas the other branch moves Southwest (S-W) toward the

69 ~~Bellingshausen Sea and in an opposite direction to the more offshore ACC. has a main trajectory~~
70 ~~from North to South (in opposition to the ACC), but from the channels around Anvers Island it~~
71 ~~moves northwards until it finds the Antarctic Coastal Current that comes from the North-East~~
72 (figure 1). The role of the APCC in the dispersal ecology ~~and evolution of Antarctic~~
73 ~~biodiversity of benthic antarctic fauna~~ has not been studied yet, but its prevalence during the
74 warmer season, when several taxa reproduce and spawn, suggests this current might be a relevant
environmental force for ~~the evolution of Antarctic biodiversity~~ larval dispersal and population
connectivity.

The bivalve *Aequiyoldia eightsi* is one of the most abundant benthic species from the shallows of
the Antarctic Peninsula (Pasotti et al. 2015). It reproduces during April-May, producing what are
thought to be lecithotrophic larvae (Lau et al. 2018), a type of non-feeding larvae with a short
planktonic stage. Given this planktonic stage, the species' dispersal potential is predicted to be
largely mediated by ocean currents (Cowen and Sponaugle 2009). The species have also been
documented in waters around Southern America, across the ACC, but recent molecular work has
suggested this might be an undescribed species despite no obvious morphological differences
(González-Wevar et al. 2019). Although the ACC (South-North direction) has been recognized
as an important potential dispersal mechanism in the Southern Ocean, currents such as the APCC
that are closer to the coast, could be more relevant for dispersal of benthic species like *A. eightsi*
in shallow waters.

Genome-wide molecular data such as that obtained from restriction site associated DNA
sequencing (RADSeq) have the power to estimate structure and patterns of gene flow at a range
of different spatial scales providing the means to detect fine scale genetic subdivision and assess
the magnitude and direction of ongoing gene flow (Xuereb et al. 2018). Understanding gene flow

and dispersal in Antarctic organisms is important because it contributes to a better understanding
of ~~population size and genetic~~ structure, ~~adaptation~~ metapopulation dynamics (Hanski 1991), and
94 resilience (Tallmon et al. 2004), all important in the context of global climate change and a
95 warming Antarctica (Barnes & Peck 2008). For instance, understanding which populations could
behave as source populations and which populations could behave as sink populations would not
only contribute to a better understanding of the oceanographic process that drive these dynamics,
but also provide valuable information for population and species conservation. In this context,
knowing which populations behave as source populations will help focussing conservation
efforts on reduced areas, making management plans more tractable. In the context of a global
climate change, identifying the currents that contribute to population connectivity may help
predicting future connectivity patterns by knowing how climate change may influence currents
and circulation patterns. In this study, we use genomic data to i) evaluate genetic structure across
the ACC, i.e. between South America and Antarctica, and within both these regions, and ii)
estimate patterns of directional gene flow to test the role of ocean currents ~~the (ACC and the~~
~~APCC)~~ on species dispersal along the West Antarctic Peninsula.

Material and Methods

*A. eightsii* is widely distributed in the Southern Ocean, but due to sampling limitations and the
need for recently collected specimens, this study focused on the southern South America and the
Western Antarctic Peninsula areas. Forty ~~onete~~ individuals of *Aequiyoldia eightsii* were
collected from five sites, two from southern South America and three ~~equally distant sites~~ from
the Western Antarctic Peninsula (figure1, Table 1). Mitochondrial DNA *COI* gene (hereafter

mtDNA) data was collected to preliminarily assess phylogeographic patterns and to confirm a
single evolutionary lineage, given that multiple lineages have been documented previously
(Gonzalez-Wevar et al. 2019). A 629 bp fragment of the cytochrome c oxidase subunit I (COI)
gene was amplified using universal primers from Folmer et al. (1994) with final concentrations
for PCR components per 25 μ L reaction as follows: 25 ng template DNA, 0.25 μ M of each
primer, 0.625 units of GoTaq DNA polymerase (Promega, Madison, WI, USA), 0.1 mM of each
dNTP, 2.5 μ L of 10 reaction buffer and 2.5 mM MgCl₂. Amplification parameters were as
follows: 95 C for 2 min followed by 35 cycles of 95 °C for 30 s, 48 °C for 30 s, and 72 °C for 90
122 s, and 72 °C for 7 min. Purification and sequencing was conducted at MACROGEN Inc. (South
Korea). Chromatograms were edited in CodonCode Aligner 8.0.2 (Dedham, MA, USA).
Sequences were imported to *BioEdit* 7.0.5.2 (Hall, 1999), aligned using the *Clustal W* algorithm
available within *BioEdit* and checked by eye. All sequences were deposited in GenBank
(accession numbers MT176643-MT176683).

Genome-wide data was obtained through the Nextera-tagmented reductively-amplified DNA
protocol (*NextRad*; SNPSaurus LLC, Eugene Oregon) of Russello et al. (2015) to provide the
adequate genetic resolution at the spatial scale analyzed (~~for further details on the molecular~~
~~protocols please refer to the Supplementary Material~~). Genomic DNA was converted into
NextRAD genotyping-by-sequencing libraries. Genomic DNA was first fragmented with Nextera
DNA FLEX reagent (Illumina, Inc), which also ligates short adapter sequences to the ends of the
fragments. The Nextera reaction was scaled for fragmenting 24 ng of genomic DNA.
Fragmented DNA was then amplified for 27 cycles at 74 degrees, with one of the primers
matching the adapter and extending 10 nucleotides into the genomic DNA with the selective
sequence GTGTAGAGCC. Thus, only fragments starting with a sequence that can be hybridized

by the selective sequence of the primer will be efficiently amplified. The nextRAD libraries were
sequenced on a HiSeq 4000 with one lane of 150 bp reads (University of Oregon).
Reads obtained from the nextRAD protocol were processed using the ipyRAD pipeline (ver.
7.30; Eaton 2014). Base calls with a quality score < 20 were converted into Ns, and any read
with > 5 Ns was discarded. Illumina adaptors and restriction sequences were removed during
filtering. Filtered reads within a sample were clustered using a threshold of 90%. Error rate and
heterozygosity were estimated from the loci clusters for each individual, and the averages were
used to establish consensus sequences. Clusters with a sequencing depth < 6 were discarded, and
only clusters with two alleles (to avoid potential paralogous loci) were retained. Consensus loci
built within samples were subsequently clustered among samples using a similarity threshold of
90% and then aligned (a maximum of eight indels allowed). Loci with heterozygous alleles
shared across more than 50% of individuals were also discarded. After data were processed, one
individual () with too few data was removed from all data sets.
Depending on the type of analysis and their sensitivity to missing data, different genomic data
sets were produced to maximize the amount of genetic information available for each analyses
by changing the levels of coverage/missing data allowed (changing the minimum number of
samples per locus for output in ipyRAD's step 7). Table 2 shows the summary statistics for data
generated by setting the minimum number of individuals per locus to ~10% (4 individuals),
~25% (10 individuals), and ~50% (20 individuals). Reads obtained from the nextRAD protocol
were processed using the ipyRAD pipeline ver. 7.30 (Eaton 2014) to obtain a series of data types
required for downstream analyses.

*Molecular analyses*

Genetic distance (p-distance) between COI sequences were calculated using the R-package
ADEGENET (Jombart 2008) to estimate levels of population differentiation. To visually display
relationships between mtDNA COI haplotypes, a haplotype network was estimated using the
median-joining network algorithm of Bandelt et al. (1999) implemented in the program *POPART*
(Leigh and Bryant 2015). In addition, mtDNA COI structure was tested calculating pairwise *Fst*
values using ~~the R-package *Adegenet*~~ADEGENET (Jombart 2008).

The genomic data generated by *NextRAD* data ~~was then~~ used to infer genetic structure through
several methods. First, relationships between each individual were estimated by building a
Maximum Likelihood (ML) tree using the program RAxML v.8.1.16 (Stamatakis 2014) with a
concatenated dataset consisting of 44,178 loci, each loci including the full sequence with
variable (SNPs) and invariable sites (matrix length= 5,973,095 base pairs). This matrix included
all loci that were present in at least 4 individuals (10 % of coverage). Including loci with missing
data has been proven desirable for phylogenetic analyses because this strategy not only increase
the amount of information available for inference (i.e. by increasing the amount of loci in the
matrix), but also because it prevent sampling biases—excluding missing data reduces the
mutational spectrum represented in the sample by disproportionally excluding loci with the
highest mutation rates (Huang & Knowles 2016). and 5,973,095 sites (including full-length loci
and all loci that presented data for at least 4 individuals). ~~T~~Given that we used the full sequence
of each locus (not only the variable sites), the GTR + Γ model of nucleotide evolution ~~was~~
~~used~~ was used as recommended in the software manual (Stamatakis 2014). Support was assessed
by 200 nonparametric bootstrap replicates, followed by a search for the best-scoring ML tree.

Subsequently, ~~T~~the software *STRUCTURE* 2.3.4 (Pritchard et al. 2000), that performs a model-
based inference of population subdivision, was used to evaluate population subdivision groups

using a single nucleotide polymorphism (SNP) data matrix of 7,781 (including loci with data for
at least ten individuals). When more than one SNP are present at a locus, the ipyRAD program
chooses randomly one of them to ensure the independence of each locus. Number of populations
to be tested (K -values) ranged from 1 to 5. Ten independent runs per K were conducted, each
with 100,000 burn-in and 300,000 MCMC iterations, using the “Admixture Model” and
“Correlated Allele Frequency Model” with default settings, except for the Lambda parameter
which was empirically estimated in a preliminary run. Results were not different using more
burn-in or MCMC iterations. *STRUCTURE HARVESTER* (Earl and vonHoldt 2012) and
*DISTRUCT* (Rosenberg 2004) were used to visualize results, and the most probable K was
chosen based on ΔK (Evanno et al. 2005). After a first round of analyses were performed,
subsequent analyses were conducted on each of the resulting clusters to determine whether any
level of substructure could be identified. In addition to the Structure analyses, a discriminant
analysis of principal components (DAPC) was conducted using the R package ADEGENET
(Jombart, 2008). This analysis transforms the data using a principal component analysis (PCA)
followed by a discriminant analysis on the uncorrelated PCA variables to produce synthetic
discriminant functions or axes that minimize within-group variation while maximizing between-
group variation (Jombart et al. 2010). First, the number of clusters were explored using the
*find.clusters* function which uses a sequential k-means clustering algorithm and Bayesian
Information Criterion to select the best-fit model (number of clusters) (Jombart et al. 2010).
Subsequently, the DAPC analysis was conducted using the estimated k value, and the actual
locations (5) as groupings for comparison, retaining in each case two PC.

Given that *A. eightsi* has a pelagic life stage, and its dispersal is likely to be influenced by
ocean currents, rates of asymmetrical migration ~~was~~ estimated ~~tested~~ using *divMigrate-*

online (Sundqvist et al. 2016) and BayesAss3-SNPS (Mussmann et al. 2019). The *divMigrate-*
online method is based on defining a hypothetical pool of migrants for a given pair of
populations and estimating an appropriate measure of genetic differentiation between each of the
two populations and the hypothetical pool. The directional genetic differentiation can then be
used to estimate the relative levels of migration between the two populations where the larger of
the two relative values indicates the population that is likely a source population, while the
smaller of the two estimates indicates the sink population. The *D* distance (Jost 2008) was used
as the measure of genetic differentiation and 200 bootstrap replicates were used for statistical
testing of assymetrical migration. ~~To calculate directional migration rates among sampling sites,~~
we used the program *divMigrate-online* (Sundqvist et al. 2016), based on the genetic distance
measures *D* (Jost 2008) and 200 bootstrap replicates for statistical support. The input data for this
analysis was the same we used for ~~the~~clusterSTRUCTURE analysis, but converted to a *genepop*
input file using *Adegenet/ADEGENET* (Jombart 2008) (see further methodological details in the
Supplementary Material). A general SW~~outh~~-NE~~orth~~ direction of gene flow (Rothera to Yelcho
and Yelcho to O'Higgins) would ~~reveal~~suggest the importance of the ACC for larval dispersal,
whereas a ~~mainly North-South gene flow~~a Yelcho to Rothera direction and a Yelcho to
O'Higgins direction would support a greater influence of the APCC as the main agent of
dispersal in the WAP. BayesAss3-SNPS (Mussmann et al. 2019), an expansion of BayesAss
(Wilson and Rannala 2003) that facilitates the use of large amounts of SNPs and automates the
fine tuning of model parameters was also used to estimate migration rates between populations.
A first round of analyses were conducted with the auto-tuning tool developed as part of
BayesAss3-SNPs, and subsequently the final analysis was conducted by running 5 replicates for
1,000,000 MCMC generations, sampling every 100 generations and discarding the first 100,000

generations as burnin. Convergence of parameter estimations were checked with Tracer v1.7.1
(Rambaut et al. 2018).

Results

15 232 *Genetic patterns from the mtDNA COI data*

Genetic differentiation in *A. eightsi* was high between South America and Antarctica. The
average number of nucleotide differences between these two regions was 38.93 (6.2 %), while
the average number of nucleotide differences within regions was 1.78 (0.28 %) and 2.28 (0.36
236 %) for South America and Antarctica, respectively. The haplotype network confirmed the high
genetic divergence between regions (figure 1), although little genetic structure was discernible
within regions. Within the West Antarctic Peninsula, only one haplogroup was found, ruling out
the presence of more than a single lineage within our samples. Analysis of genetic structure
based on *Fst* (Table S1, electronic Supplementary Material) showed, unsurprisingly, high genetic
differentiation between South America and Antarctica. More surprising, however, was the
finding of statistically significant differentiation between some Antarctic populations. There was
significant differentiation between O'Higgins Base and the other two Antarctic sites, Yelcho
(*Fst*= 0.36, p= 0.012) and Rothera (*Fst*= 0.469, p= 0.008), but no significant differentiation was
found between Yelcho and Rothera (*Fst*= 0.072, p= 0.304) and neither was found between the
two South American populations (*Fst*= 0.142, p= 0.131).

*Genomic data and ~~G~~genetic structure revealed by the NextRAD data*

The total number of loci found by ipyRAD was 352,263. After the filtering process and quality
controls, the total amount of loci was 297,417. As with many NexGen data sets, which are
characterized by moderate to high levels of missing data (Huang and Knowles 2016; Tripp et al.
2017; Hodel et al. 2017), most of these loci were present only for a few number of individuals.
For instance, most of these loci (253,239) were present at 3 or fewer samples. Subsequently,
datasets obtained with the constraint of having loci represented for at least 4, 10, and 20
individuals produced a total of 44,178, 8808, and 941 loci, respectively. Locus length varied
from 16 bp to 266 bp, with a mean locus length of 135 bp. Of these loci, 5090 were invariable,
whereas 39088 were variable. The number of variable sites (SNPs) per locus varied from 1 to 19,
with an average of 5.4 SNPs per locus.

ML phylogenetic relationships showed strong phylogeographic structure, with clades strongly
reflecting geography (figure 2A). Support for the different clades varied from very high at the
Rothera site (bootstrap support= 100) to weak in the Puerto Williams clade (bootstrap support=
53). Antarctica and South America were recovered as the two major clades, with all populations
within these regions being recovered as reciprocally monophyletic.

The *STRUCTURE* analyses confirmed the strong genetic differentiation between the South
American and Antarctic regions. For the entire dataset, including all samples in a single analysis,
the highest supported K (as suggested by the Evanno test) was $K= 2$, matching exactly the
Antarctic and South American regions. After rerunning *STRUCTURE* independently for each
region, the results did not support genetic substructure in South America. On the other hand, the
analysis for the Antarctic region supported the existence of two clusters ($K= 2$) with varying
degrees of admixture. One cluster was mainly associated with Rothera, while the second cluster

was mainly present in Yelcho. The northernmost site, O'Higgins Base, presented individuals with
different levels of admixture from both clusters.

The DAPC results were broadly consistent with the *Structure* results (figure 3). The three first
PC axes explained, cumulatively, 24.86 % of the total variance, with the first, second and third
PC axes explaining 21.91 %, 3.45 %, and 3.38 % of the total variance, respectively (PC1 vs PC2
plot and other details can be seen in figure S2-S4). The DAPC showed the Antarctic and South
American regions well differentiated along the PC1 axis and very little differentiation was evident
along the PC2 axis. Within the Antarctic sites, Rothera was the most differentiated, whereas the
O'Higgins Base and Borguen Bay sites showed some differentiation, but a greater overlap. The
Magellan Strait and Puerto Williams sites in the South American region showed moderate
differentiation. The BIC supported $k=2$ clusters that best described the data (figure S3),
consistent with the South American and Antarctic regions, although two individuals from the
South American region showed admixed memberships (PW_04 and EM_06) with the Antarctic
locality Yelcho, a result that may be influenced by a high level of missing data for these
individuals which, coincidentally, presented the lowest number of loci in the data set.

~~The *STRUCTURE* analyses confirmed the strong genetic differentiation between the two regions.~~
~~For the entire dataset, including all samples in a single analysis, the highest supported K (as~~
~~suggested by the Evanno test) was $K=2$, matching the Antarctic and South American regions.~~
~~After rerunning *STRUCTURE* independently for each region, the results did not support genetic~~
~~substructure in South America. On the other hand, the analysis for the Antarctic region supported~~
~~the existence of two clusters ($K=2$) with varying degrees of admixture. One cluster was mainly~~
~~associated with Rothera, while the second cluster was mainly present in Yelcho. The~~

northernmost site, O'Higgins Base, presented individuals with different levels of admixture from
both clusters.

The analysis of directional gene flow revealed an important asymmetry in the migration rates
between populations in the WAP, ~~with a strong tendency to flow in a North-South direction~~
(figure 2B). A moderate, but significantly higher migration rate was detected from Yelcho to
Rothera ($D= 0.30$) as well as from Yelcho to O'Higgins Base ($D= 0.17$) in agreement with a role
for the APCC on larval dispersal. Strikingly, tThe highest rate of migration occurred ~~between~~from
O'Higgins Base ~~and~~to Rothera ($D= 0.69$), ~~followed by the rate of migration from Yelcho to~~
Rothera ($D= 0.30$). Some reduced migration was also detected from Yelcho to Base O'Higgins
($D= 0.17$). No asymmetrical migration was detected in the direction of the Yelcho population.
High migration rates were inferred between ~~Asymmetric gene flow between~~ the two South
American sites, ~~although no~~ ~~was not~~ significant ~~asymmetrical migration was found~~ (figure S1
and Table S2, Electronic Supplementary Material).

The BatesAss analysis produced qualitatively similar results to the divMigrate-online analysis.
The mean migration rate from Yelcho to Rothera (0.024) was slightly higher than from Rothera
to Yelcho (0.022) and the mean migration rate from Yelcho to O'Higgins Base (0.026) was
slightly higher than from O'Higgins Base to Yelcho (0.022). The migration rate from O'Higgins
Base to Rothera was 0.024, whereas the migrtaiion rate from Rothera to O'Higgins Base was
0.026. However, confidence intervals were very wide and overlapping indicating no statistical
support for asymmetrical migration.

Discussion

~~We found no evidence of significant migration between Antarctica and South America in the~~
~~bivalve *Aequiyoldia eightsi*. Indeed, the deep divergence between these regions found by~~
~~previous studies (e.g. Poulin et al. 2014; González-Wevar et al. 2019) was strongly supported by~~
~~both the mtDNA and the genome-wide NextRAD data sets, confirming the Drake passage is an~~
~~effective barrier for the species.~~ Genome-wide markers have been recently used to support
ongoing gene flow across the Drake passage in the brittle star *Astrota agassizii* (Galaska et al.
2017a), in contrast to the evidence of no connectivity found earlier for the species based only on
a few markers (Hunter and Halanych, 2008). There is also recent evidence, based on SNP
markers, that marine benthos, indeed brittle stars, have northwards gene flow against the ACC
from the Southern Ocean to South America (Sands et al. 2015). Thus, different sources of
genetic information may can often provide, ~~in some cases~~, new insights about the processes
driving patterns of spatial genetic diversity. In this study, we found no evidence of significant
migration between Antarctica and South American populations of the bivalve *Aequiyoldia*
*eightsi* confirming the results found by previous studies based on fewer markers (e.g. Poulin et
al. 2014; González-Wevar et al. 2019). However, in *A. eightsi* the lack of ongoing gene flow
across the Drake passage found in the present study using a dataset of 7,781 SNPs suggests that
migration is very limited across this gap between continents. Although the results of the DAPC
analysis yielded conflicting results by showing two individuals from the South American region
with membership probabilities mixed with Antarctic populations, all other analyses
(phylogenetics and Structure) showed a strong differentiation between regions. Thus, it is likely
that the conflicting evidence correspond to a methodological artifact due to the presence of
missing data. Although the impact of missing data has not been thoroughly studied in the context
of genetics, it is known that missing data can severely impact PCA performance, specially when

missing data is not distributed randomly (Dray and Josse 2015). In fact, in our data set the two
individuals that presented the conflicting evidence (EM_07 and PW_04) were also the ones that
had the lowest amount of loci (table S5 in the Supplementary Material) suggesting that the lack
of information in these individuals may had attracted their position in the PCA to the center of
the plot, putting them closer to the Antarctic cluster.

Within regions, genetic structure was onlymore evident in the WAP. Both mtDNACOI and
NextRADgenomic data evidenced highsome degree of genetic structure between the O'Higgins
Base and the other two southern sites Yelcho and Rothera, but low genetic structure between
Yelcho and Rotherapopulations, with Rothera identified as more distinct by both Structure and
DAPC. No such differentiation was seen with the COI data, perhaps due to the lower resolution
of the marker.

When the direction of gene flow was estimated, a strong pattern of migration from North to
South emergedWe found evidence of asymmetrical migration between Antarctic populations that
was consistent with the circulation patterns of the APCC (figure 2B), suggesting there is a
strong prevalence of dispersal southwards. A predominant migration from Yelcho to O'Higgins
Base and from Yelcho to Rothera matches the circulation of the APCC in this area, that moves in
opposite directions from around Yelcho (Anvers Island). From Anvers Island, one flow goes in a
Southwest direction to the Margarite Bay where the Rothera Base is located, whereas the other
flow moves in a Northeast direction until it joins the Antarctic Coastal Current (CC) that comes
from the Weddell Sea and through the Bransfield Strait (Moffat & Meredith 2018). More
puzzling, however, is the asymmetrical migration inferred from O'Higgins Base to Rothera
because near-coast southward migration would be prevented by the northeast flow of the APCC.
Currents in this area are indeed very complex, and most models depicting trends and flow

directions are strong simplifications of the complexity of the currents. There is evidence, that
ocean circulation in the WAP, can be time dependant and, at times, current directions can vary
during some periods of time (Moffat & Meredith 2018; Renner et al. 2012). Renner et al. (2012),
using an ice-ocean modelling approach and empirical data from near-surface drifters released in
the Weddell Sea near the tip of the Antarctic Peninsula, found that although most of
connectivity goes NE across the Scotia Sea towards South Georgia, a significant portion goes
initially in a West direction around the tip of the Antarctic Peninsula and then southward to the
Bellingshausen Sea. This ocean circulation fits well with the gene flow pattern observed in *A.*
*eightsii* because it would mean that larvae transportation from the tip of the Antarctic Peninsula
to Rothera can occur following this southward current at some distance from the coast, avoiding
the near-coast, northeast-flow branch of the APCC. Based on the genetic results we suggest that,
although current systems in this area are complex, movements from the tip of the Peninsula
toward the Bellingshausen Sea (particularly some offshore events) area frequent enough, an
comparatively more predominant than viceversa, to facilitate larval dispersal.

Although there is no direct evidence on the larval type of *A. eightsi*, species in the subclass
Protobranchia are known to exhibit either pelagic lecithotrophy or brooding larval development
(Zardus 2002). Lau et al. (2018), based on monthly examinations of reproductive tissues of *A.*
*eightsi*, suggested a pelagic lecithotrophic larval development for the species, a hypothesis that is
supported by our results, given the correlation between the local ocean current (APCC) direction
and the migration patterns observed in *A. eightsi*. The APCC has a strong seasonality, being
active during November-June, and disappearing during July-October in concordance with
freshwater input cycles (Moffat & Meredith 2018). The peak of reproduction and spawning of *A.*
*eightsi* observed between April and May (Lau et al. 2018) suggests the dispersal of the species is

~~likely facilitated by the APCC. Asymmetrical migration from Yelcho to O'Higgins Base was also~~
~~consistent with the direction of the APCC in this area, which moves northwards until it joins the~~
~~Antarctic Coastal Current (CC) that comes from the North East. Perhaps, this migration~~
~~northwards explain why the highest migration rate was found between O'Higgins Base to~~
~~Rothera and not from Yelcho to Rothera, despite the closer geographical distance between~~
~~Yelcho and Rothera.~~
~~The APCC has a strong seasonality, being active during November-June, and disappearing~~
~~during July-October in concordance with freshwater input cycles (Moffat & Meredith 2018). The~~
~~peak of reproduction and spawning of *A. eightsi* observed between April and May (Lau et al.~~
~~2018) suggests the dispersal of the species is facilitated by the APCC.~~

The role of the ACC on species dispersal has been well documented for several taxa, but
~~little much less~~ is known of the importance of other, more localized (and closer to the coast)
currents in this regards. As far as we are aware, this is the first study providing evidence of the
role of the APCC on the dispersal of a marine invertebrate species. ~~Our data also suggest that~~
~~more complex current dynamics may also be important for transporting larva from the tip of the~~
~~Antarctic Peninsula to the Bellingshausen Sea area (Renner et al. 2012), although further~~
~~genetic, ecological and oceanographic studies are needed to fully understand these dynamics.~~
~~Future-sAdditional~~ studies comparing genetic patterns between species with different larval
development strategies will provide ~~a better understanding further insights of about~~ the
importance of ~~this and other~~ Southern Ocean currents to the dispersal and distribution of
Antarctic benthic organisms and ~~its their~~ role on the evolution of Antarctic biodiversity.

Ethics

This study did not require approval from an ethics committee.

Data accessibility

All data used in this study, including the mtDNA COI sequences and NextRAD genomic data

input files are available at Dryad

(https://datadryad.org/stash/share/u30ZZTnVWdldwbDBChsJ1zuni0TVuwRFPVwcv_bQwO8)

(Munoz-Ramirez 2019). COI sequences are also available at GenBank through accession

numbers MT176643-MT176683.

Authors' contributions

CPM-R conducted the lab work, genomic analyses and drafted the manuscript. DKAB, CJS, JS,

AR-G, SAM, LC, MM, and AB ~~carried out~~ contributed to the fieldwork sampling, provided

inputs for the sampling design and drafted and edited the manuscript (or significant parts of it).

All authors gave final approval for publication and agree to be held accountable for the content

herein.

Competing interests

We have no competing interests.

Funding

This research was supported by an international collaborative project (ICEBERGs), a CONICYT-

NERC grant (223449) to AB and project FONDECYT postdoctorado (3180331) to CPM-R.

Acknowledgments

We thank the crew of the James Clark Ross vessel for their valuable assistance in the field and to

INACH for granting the necessary permits.

1
2
3 426
4
56 427 References

Allcock AL, Barratt I, Eléaume M, et al (2011) Cryptic speciation and the circumpolarity
debate: A case study on endemic Southern Ocean octopuses using the COI barcode of life. Deep
Res Part II Top Stud Oceanogr 58:242–249. doi: 10.1016/j.dsr2.2010.05.016

Bandelt HJ, Forster P, Röhl A (1999) Median-joining networks for inferring intraspecific
phylogenies. Mol Biol Evol 16:37–48. doi: 10.1093/molbev/msl151

Barnes DKA, Peck LS (2008) Vulnerability of Antarctic shelf biodiversity to predicted regional
warming. Clim Res 37:149–163. doi: 10.3354/cr00760

Clarke, A., Crame, J., 1989. The origin of the Southern Ocean marine fauna. J. Geol. Soc. Lond.
47, 253–268.

Clarke A, Barnes DKA, Hodgson DA 2005 How isolated is Antarctica? Trends Ecol Evol 20:1–
3. doi: 10.1016/j.tree.2004.10.004

Cowen RK, Sponaugle S (2009) Larval Dispersal and Marine Population Connectivity. Ann Rev
Mar Sci 1:443–466. doi: 10.1146/annurev.marine.010908.163757

Dray S, Josse J (2015) Principal component analysis with missing values: a comparative survey
of methods. Plant Ecol 216:657–667. doi: 10.1007/s11258-014-0406-z

Eaton, D. A. R. 2014. PyRAD: assembly of de novo RADseq loci for phylogenetic analyses.
Bioinformatics 30:1844–1849.

Evanno G, Regnaut S, Goudet J (2005) Detecting the number of clusters of individuals using the
software structure: a simulation study. *Mol Ecol* 14:2611–2620. doi: 10.1111/j.1365-
294X.2005.02553.x
Folmer O, Black M, Hoeh W, et al (1994) DNA primers for amplification of mitochondrial
cytochrome c oxidase subunit I from diverse metazoan invertebrates. *Mol Mar Biol*
*Biotechnol* 3:294–299. doi: 10.1371/journal.pone.0013102
Galaska MP, Sands CJ, Santos SR, et al (2017) Crossing the divide: Admixture across the
antarctic polar front revealed by the brittle star *astrotoma agassizii*. *Biol Bull* 232:198–211.
doi: 10.1086/693460
Galaska MP, Sands CJ, Santos SR, et al (2017) Geographic structure in the Southern Ocean
circumpolar brittle star *Ophionotus victoriae* (Ophiuridae) revealed from mtDNA and
single-nucleotide polymorphism data. *Ecol Evol* 7:475–485. doi: 10.1002/ece3.2617
González-Wevar CA, Hüne M, Segovia NI, et al (2017) Following the Antarctic Circumpolar
Current: patterns and processes in the biogeography of the limpet *Nacella* (Mollusca:
Patellogastropoda) across the Southern Ocean. *J Biogeogr* 44:861–874. doi:
10.1111/jbi.12908
González-Wevar CA, Gérard K, Rosenfeld S, et al (2019) Cryptic speciation in Southern Ocean
*Aequiyoldia eightsi* (Jay, 1839): Mio-Pliocene trans-Drake Passage separation and
diversification. *Prog Oceanogr* 174:44–54. doi: 10.1016/j.pocean.2018.09.004
Hanski I. 1991. Single-species metapopulation dynamics: concepts, models and observations.
Biol J Linn Soc 42:17-38. doi:10.1111/j.1095-8312.1991.tb00549.x.

Hemery LG, Eléaume M, Roussel V, et al (2012) Comprehensive sampling reveals
circumpolarity and sympatry in seven mitochondrial lineages of the Southern Ocean crinoid
species *Promachocrinus kerguelensis* (Echinodermata). *Mol Ecol* 21:2502–2518. doi:
10.1111/j.1365-294X.2012.05512.x
Huang H, Lacey Knowles L (2016) Unforeseen consequences of excluding missing data from
next-generation sequences: Simulation study of rad sequences. *Syst Biol* 65:357–365. doi:
10.1093/sysbio/syu046
Hodel, R.G.J., Chen, S., Payton, A.C. et al. Adding loci improves phylogeographic resolution in
red mangroves despite increased missing data: comparing microsatellites and RAD-Seq and
[investigating loci filtering. *Sci Rep* 7, 17598 \(2017\). \[16810-7\]\(https://doi.org/10.1038/s41598-017-
Poulin E, González-Wevar C, Díaz A, et al (2014) Divergence between Antarctic and South
American marine invertebrates: What molecular biology tells us about Scotia Arc
geodynamics and the intensification of the Antarctic Circumpolar Current. *Glob Planet*
*Change* 123:392–399. doi: [10.1016/j.gloplacha.2014.07.017](https://doi.org/10.1016/j.gloplacha.2014.07.017)
Pasotti F, Manini E, Giovannelli D et al (2015a) Antarctic shallow water benthos in an area of
recent rapid glacier retreat. *Mar Ecol* 36:716–733.
Pritchard, J. K., M. Stephens, and P. Donnelly. 2000. Inference of population structure using
multilocus genotype data. *Genetics*
155:945–959.
Rambaut A, Drummond AJ, Xie D, Baele G and Suchard MA (2018) Posterior summarisation in
Bayesian phylogenetics using Tracer 1.7. *Systematic Biology*. syy032.
[doi:10.1093/sysbio/syy032](https://doi.org/10.1093/sysbio/syy032)
Raupach MJ, Thatje S, Dambach J, et al (2010) Genetic homogeneity and circum-Antarctic
distribution of two benthic shrimp species of the Southern Ocean, *Chorismus antarcticus*
and *Nematocarcinus lanceopes*. *Mar Biol* 157:1783–1797. doi: [10.1007/s00227-010-1451-3](https://doi.org/10.1007/s00227-010-1451-3)

Rosenberg NA (2004) DISTRUCT: A program for the graphical display of population structure.
Mol Ecol Notes. doi: 10.1046/j.1471-8286.2003.00566.x
Rozas J, Ferrer-Mata A, Sanchez-DelBarrio JC, et al (2017) DnaSP 6: DNA sequence
polymorphism analysis of large data sets. Mol Biol Evol. doi: 10.1093/molbev/msx248
Russello MA, Waterhouse MD, Etter PD, Johnson EA (2015) From promise to practice: Pairing
non-invasive sampling with genomics in conservation. PeerJ 2015:1–18. doi:
10.7717/peerj.1106
Sands CJ, O’Hara TD, Barnes DKA, Martín-Ledo R (2015) Against the flow: Evidence of
multiple recent invasions of warmer continental shelf waters by a Southern Ocean brittle
star. Front Ecol Evol 3:1–13. doi: 10.3389/fevo.2015.00063
Stamatakis A (2014) RAxML version 8: A tool for phylogenetic analysis and post-analysis of
large phylogenies. Bioinformatics 30:1312–1313. doi: 10.1093/bioinformatics/btu033
Sundqvist L, Keenan K, Zackrisson M, et al (2016) Directional genetic differentiation and
relative migration. Ecol Evol 6:3461–3475. doi: 10.1002/ece3.2096
Tallmon DA, Luikart G, Waples RS. 2004. The alluring simplicity and complex reality of
genetic rescue. Trends Ecol Evol 19:489-96. doi:10.1016/j.tree.2004.07.003.
Thatje S, Hillenbrand CD, Larter R (2005) On the origin of Antarctic marine benthic community
structure. Trends Ecol Evol 20:534–540. doi: 10.1016/j.tree.2005.07.010
Tripp, E.A., Tsai, Y.H.E., Zhuang, Y. and Dexter, K.G., 2017. RAD seq dataset with 90%
missing data fully resolves recent radiation of Petalidium (Acanthaceae) in the ultra-arid
deserts of Namibia. Ecology and Evolution, 7(19), pp.7920-7936.

Xing S, Hou X, Aldahan A, et al (2017) Water Circulation and Marine Environment in the
Antarctic Traced by Speciation of ¹²⁹I and ¹²⁷I. Sci Rep 7:1–9. doi: 10.1038/s41598-017-
07765-w
Xuereb A, Benestan L, Normandeau É, et al (2018) Asymmetric oceanographic processes
mediate connectivity and population genetic structure, as revealed by RADseq, in a highly
dispersive marine invertebrate (*Parastichopus californicus*). Mol Ecol 27:2347–2364. doi:
10.1111/mec.14589
Zardus JD (2002) Protobranch bivalves. Adv Mar Biol 42:1–65

Appendix B

Dear Editor,

We appreciate the new comments and suggestions by the two reviewers. Below we provide a point-by-point response to all the questions asked by the reviewers (in blue text) and have made the corrections where appropriate.

Associate Editor's comments (Dr Michael Tobler):

Comments to the Author:

We have received the feedback from two reviewers, both of which provided a generally positive assessment of the manuscript. Before the manuscript can be accepted, however, the authors need to address some substantial issues with their analyses. In addition, both reviewers provided some minor suggestions that should be addressed before the manuscript is acceptable for publication in RSOS.

Reviewers' Comments to Author:

Reviewer: 1

Comments to the Author(s)

The manuscript intitle "Gene flow in the Antarctic bivalve *Aequiyoldia eightsii* (Jay, 1839) suggests a role for the Antarctic Peninsula Coastal Current in larval dispersal", which was previously submitted to Biology Letters and reviewed by me, incorporated most of the suggestions, making changes in all sections. The writing is clearer, the methodology more detailed, new analyses were included, and the discussion became more robust. As previously mentioned, the manuscript is interesting, technically sound and conducted in a region that is difficult to access and relatively poor explored in phylogeographic and populational genetic terms. Thus, I consider that the manuscript can be accepted for publication.

Minor corrections/suggestions:

Line 109 – Change "A. eightsii" to "*Aequiyoldia eightsii*".

Response: Fixed.

Line 122 – Change "and 72° C for 7 min" to "and a final extension of 72° C for 7 min".

Response: Fixed.

Line 149 – There is an empty parentheses ().

Response: Fixed.

Line 153 – Table 2 was not found. Please check.

Response: We are sorry for the confusion. This Table was finally moved to the supplementary material, named Table S1. We forgot to change it in the version with track changes as it was changed at the very last minute directly in the clean version.

Line 240 and 241 – Perhaps the expressions "unsurprisingly" and "more surprising" can be replaced by "as expected" and "not expected".

Response: Fixed.

Reviewer: 2

Comments to the Author(s)

I have revised the second version of this manuscript, and I commend the authors on their efforts to improve it. However, I am still uneasy with the level of missing data being allowed in the population structure and migration analyses. I see that some analyses with less missing data were included in supplementary material, but they are not referred to in the main body of text, and it is not clear if migration tests were performed using that improved dataset. This will need to be clarified before acceptance.

Response: We thank this suggestion. We understand that there is a delicate balance between removing missing data and reducing number of loci versus allowing missing data but also including more loci in the analyses. Some research has been done regarding this issue and we have paid attention to it given that we chose a protocol for generating genetic data that is affordable, but that can contain high levels of missing data (common for RADSeq protocols). Some other protocols can reduce missing data (e.g. SNPs panels or UCEs), but at the cost of lower genetic variability or high monetary costs or that can only be used for non-model organisms.

In particular we have learnt that missing data should not be removed from phylogenetic analysis like our RAxML analysis [1]. Also, we learnt that missing data do not adversely impact assignment analyses such as those from STRUCTURE. The extract below is from a paper [2] that specifically tested the impact of missing data on population assignments in the program STRUCTURE, which we used for our analyses:

“Presence of mild to considerable extent of missing data (Additional file 1: Table S4) also did not adversely affect the assignment scores in Structure and most individuals were ascribed group memberships in agreement with the microsatellite dataset. However one individual (CA002) considered as genetic intermediate (microsatellite data) was consistently assigned to *C. brachyotis* based on the SNP analysis”

We chose to remove only excessive amounts of missing data, maintaining a greater amount of information in terms of number of loci usable for analyses, which sometimes meant an order of magnitude more loci, and consequently, more power for analyses. Nevertheless, and taking the reviewer’s suggestion, we have mentioned where appropriate the characteristics of the data used and discussed about potential limitations.

1. Chattopadhyay B, Garg KM, Ramakrishnan U. 2014 Effect of diversity and missing data on genetic assignment with RAD-Seq markers. BMC Res. Notes 7, 4–6. (doi:10.1186/1756-0500-7-841)

2. Huang H, Lacey Knowles L. 2016 Unforeseen consequences of excluding missing data from next-generation sequences: Simulation study of rad sequences. Syst. Biol. 65, 357–365. (doi:10.1093/sysbio/syu046)

Furthermore, the results presented are somewhat contradictory between analyses, but that is not discussed in detail.

Response: We realized that there were a few instances where contradictory results were not fully discussed. These were now discussed in more detail, providing potential explanations or recognizing the answers were elusive due to the complexity of the system.

Major changes were made in the second paragraph of the discussion section and we have also reworded additional sentences in the first paragraph of the Results section.

Therefore, I am recommending major revisions. All my comments can be found in the attached pdf (RSOS-200603_Proof_hi_reviewed.pdf).

I have two other comments here:

What is sample BB2_G1_3? This code does not match the other samples codes

Response: This is a sample from Yelcho that was collected a year later than the other samples, and named differently by another researcher.

Please revise the supplementary material file, as there is a table without contents, and some of the figures do not say which dataset was used to generate them.

Response: Thank you for checking this out. The empty table was a table that was moved upwards in the document, but for some reason, the frame was kept in place. Removed now.

We have added the information about data sets used, making reference to the input files described in the supplementary material.

We have fixed all other minor issues from the provided pdf.

Appendix C

Dear Editor,

We are very glad for the positive review and decision. We have now fixed all the minor issues suggested by reviewers. Below we provide a point-by-point response.

Reviewer comments to Author:

Reviewer: 1

Comments to the Author(s)

After this new round of review, all my questions were resolved, so the article can be accepted without further corrections.

Reviewer: 2

Comments to the Author(s)

I commend the authors for a good review and I think the manuscript is ready for acceptance, pending some (very) minor changes for typos and consistency.

Minor changes:

- page 7 line 17: change degrees for the scientific notation (oC)

Replay: Fixed

- page 7 line 62: change structure for STRUCTURE

Reply: Fixed

- revise notation of numbers and decimal points throughout to ensure consistency (some times it has two decimal points, some times three; thousands are some times separated by commas, others not)

Replay:

- page 8 lines 57-59: I'm not sure what loci the authors are referring to in this section.

Replay: Thank you. After a thorough revision we fixed the following:

- page x, line x, we changed 1000 bootstrap replicates to 1,000 bootstrap replicates.

- page x2, line x2, we changed 6.2 % to 6.19 %

- page x3, line x3, we changed $F_{st} = 0.36$ to $F_{st} = 0.360$

- page x4, line x4, we changed 8808 to 8,808

- page x5, line x5, we changed 5090 to 5,090

- page x6, line x6, we changed 39088 to 39,088

- page x7, line x7, we changed 39088 to 39,088

- page x8, line x8, we changed 39088 to 39,088

- page 9: cite figure 2 when mentioning STRUCTURE results

Replay: fixed.

- page 10 lines 10-18: no reference to mtDNA FST results (and how they do not really match nuclear SNPs FST results)

Replay: We have provided now reference to the Fst results with a potential reason for discrepancies between data sets.

- Figures: consider be consistent with colours for sampling sites. E.g. O'Higgins is represented in yellow in Fig1 but in green in Fig 3 (Yelcho is yellow in this figure)

Replay: Colors from figure 1 were changed to be consistent with figure 3.

Other minor changes were made to make text more clear, including leyend of Table 1 and some italicized text was changed to normal text for software or analysis names.